# Somatic mTOR mutation in clonally expanded T lymphocytes associated with chronic graft versus host disease

Daehong Kim [1,2,13], Giljun Park [1,2,13], Jani Huuhtanen [1,2], Sofie Lundgren[1,2], Rajiv K. Khajuria[1], Ana M. Hurtado [3], Cecilia Muñoz-Calleja[4], Laura Cardeñoso[5], Valle Gómez-García de Soria[6], Tzu Hua Chen-Liang [3], Samuli Eldfors [7], Pekka Ellonen[7], Sari Hannula[7], Matti Kankainen[1,2,8,9], Oscar Bruck [1,2,9], Anna Kreutzman[1,2], Urpu Salmenniemi[10], Tapio Lönnberg[11], Andrés Jerez[3], Maija Itälä-Remes[10], Mikko Myllymäki [1,2], Mikko A. I. Keränen[1,2,12] & Satu Mustjoki [1,2,9]✉

Graft versus host disease (GvHD) is the main complication of allogeneic hematopoietic stem cell transplantation (HSCT). Here we report studies of a patient with chronic GvHD (cGvHD) carrying persistent CD4$^+$ T cell clonal expansion harboring somatic *mTOR*, *NFKB2*, and *TLR2* mutations. In the screening cohort (n = 134), we detect the *mTOR P2229R* kinase domain mutation in two additional cGvHD patients, but not in healthy or HSCT patients without cGvHD. Functional analyses of the *mTOR* mutation indicate a gain-of-function alteration and activation of both mTORC1 and mTORC2 signaling pathways, leading to increased cell proliferation and decreased apoptosis. Single-cell RNA sequencing and real-time impedance measurements support increased cytotoxicity of mutated CD4$^+$ T cells. High throughput drug-sensitivity testing suggests that mutations induce resistance to mTOR inhibitors, but increase sensitivity for HSP90 inhibitors. Our findings imply that somatic mutations may contribute to aberrant T cell proliferations and persistent immune activation in cGvHD, thereby paving the way for targeted therapies.

[1] Hematology Research Unit Helsinki, Helsinki University Hospital Comprehensive Cancer Center, 00290 Helsinki, Finland. [2] Translational Immunology Research Program and Department of Clinical Chemistry and Hematology, University of Helsinki, 00014 Helsinki, Finland. [3] Hematology and Medical Oncology Department, Hospital Morales Meseguer. Centro Regional de Hemodonación, Universidad de Murcia, IMIB, 30008 Murcia, Spain. [4] Department of Immunology, Hospital Universitario de La Princesa, Instituto de Investigación Sanitaria Princesa, 28006 Madrid, Spain. [5] Department of Microbiology, Hospital Universitario de La Princesa, Instituto de Investigación Sanitaria Princesa, 28006 Madrid, Spain. [6] Department of Hematology, Hospital Universitario de La Princesa, Instituto de Investigación Sanitaria Princesa, 28006 Madrid, Spain. [7] Institute of Molecular Medicine Finland (FIMM), University of Helsinki, 00014 Helsinki, Finland. [8] Department of Medical and Clinical Genetics, University of Helsinki and Helsinki University Hospital, 00014 Helsinki, Finland. [9] iCAN Digital Precision Cancer Medicine, University of Helsinki and Helsinki University Hospital, 00014 Helsinki, Finland. [10] Stem Cell transplantation unit, Turku University Hospital, 20521 Turku, Finland. [11] Turku Bioscience Centre, University of Turku and Åbo Akademi University, 20520 Turku, Finland. [12] Department of Hematology, Helsinki University Hospital Comprehensive Cancer Center, 00029 Helsinki, Finland. [13] These authors contributed equally: Daehong Kim, Giljun Park. ✉email: satu.mustjoki@helsinki.fi

Graft versus host disease (GvHD) is a major source of morbidity and mortality after allogeneic hematopoietic stem cell transplantation (allo-HSCT)[1]. Chronic GvHD (cGvHD) occurs after day 100 post-transplantation and develops in 30–70% of allo-HSCT recipients[2]. The 5-year mortality rate is 30–50% and can be attributable to immune dysregulation and infections[3]. Affected patients frequently need immunosuppressive treatment for years or even for a lifetime.

The pathophysiology of cGvHD is multifactorial. Engrafted donor CD4[+] T cells contribute to tissue injury, inflammation, and late aberrant tissue repair and fibrosis, making them candidate therapy targets in cGvHD[4]. In many immune system mediated disorders, such as cGvHD, the immune homeostasis is disturbed, and the enormous variability of T cell receptor (TCR) repertoire is diminished. In some patients, clonal expansions comprising up to 20–40% of all T cells can exist[5–7]. While somatic mutations have been identified in clonally expanded T cells in human diseases, such as in large granular lymphocyte (LGL) leukemia[8–10], aplastic anemia (AA)[11], and rheumatoid arthritis[12], the prevalence and impact of this phenomenon has not been investigated in patients with cGvHD.

Here, we hypothesize that in cGvHD activated T cells may acquire somatic mutations due to constant immune-system activation and proliferation. Such mutations might lead to functional and survival advantage, resulting in clonal expansion and further activation of the immune response. To explore this theory, we purify CD4[+] and CD8[+] lymphocytes from an index patient suffering from persistent cGvHD despite long-term immunosuppressive therapies and sequence them with a custom deep-sequencing panel consisting of immunity and inflammation-related genes. We study the discovered mutations in a validation cohort of 134 cGvHD patients. Subsequently, we evaluate the functional consequences of the discovered mutations in vitro. Finally, we verify these observations in patient cells using drug sensitivity screening and unbiased transcriptome-wide single-cell RNA-sequencing (scRNA-seq) paired with TCR alpha and beta sequencing (TCRαβ-seq) analysis and functional cytotoxicity assays.

## Results

**Clinical characteristics of the index cGvHD patient**. The index patient was a 56-year-old male, who was diagnosed with chronic phase chronic myeloid leukemia in 1999. The patient received an allogeneic bone marrow transplantation in 2000. Since the beginning of 2001, the patient suffered from cGvHD affecting his liver, eyes, nails, and skin. The immunosuppression was continuously adjusted according to the clinical presentation of the cGvHD. The clinical status and treatment history are described in detail in Supplementary Fig. 1 and Supplementary results.

**Immunophenotype and TCR repertoire of T cells**. During the first sampling in 2013, the patient was on mycophenolate mofetil therapy. TCR repertoire was initially analyzed with a flow cytometry-based assay using a panel of TCR Vβ-specific antibodies with the sample obtained in 2015. A large clonal Vβ20+ expansion was noted among CD4[+] T cells (Fig. 1a, b). The Vβ20+ clone constituted 52.9% of total CD4[+] T cells, and it increased over time (Fig. 1a, c). Smaller clonal expansions were detected in total CD8[+] T cells, but not the Vβ20+ expansion (Fig. 1a, Supplementary Fig. 2). To assess the repertoire in more detail, FACS-sorted CD4[+] Vβ20+, CD4[+] Vβ20−, and CD8[+] T cells were further analyzed by a TCRβ deep-sequencing assay[12], which confirmed the TCRBV30-01 expansion (corresponding to the Vβ20+ expansion observed by flow cytometry) in the CD4[+]Vβ20+ fraction (66.2% of sorted cells) (Fig. 1d). TCRβ

sequencing of CD8[+] T cells revealed two clones, TCRBV07-09 (16.1%) and TCRBV28-01 (17.9%) (Fig. 1d).

During an exacerbation of sclerodermatous skin lesions in 2015, 59% of peripheral blood leukocytes were T cells, 5% B cells, and 35% NK cells (Supplementary Fig. 2a). CD3[+] T cells were composed of CD4[+] (59.3%), CD4[+]CD8[+] (11.3%), and CD8[+] T cells (12.6%) (Supplementary Fig. 2b). An increased number of CD4[+] effector memory (EM, 75.0%) and terminally differentiated effector memory (TEMRA) cells (17.4%) was found together with a decreased number of CD4[+] central memory (CM) cells (6.2%) when compared with the sibling HSCT donor's CD4[+] T cell pool (59.6% EM, 5.0% TEMRA, and 19.9% CM cells) (Fig. 1e). In the CD8[+] T cell pool, increased amount of TEMRA cells was noted (79.9% of CD8[+] T cells). The proportion of cells positive for cytotoxic enzyme granzyme B (GrB) was notably high both among CD4[+] and CD8[+] T cells (46% and 87%, respectively, Fig. 1f).

**Somatic mutations in the expanded CD4[+] T cell population**. To screen for somatic mutations, a customized immunity and inflammation-related gene sequencing panel (immunogene panel)[12,13] was applied to immunomagnetic bead-separated blood CD4[+] and CD8[+] T cells that were obtained from the index patient in 2013. The median target gene coverage for the panel was 152 in CD4[+] and 160 for CD8[+] T cells. In total, 14 candidate putative somatic mutations were discovered within the CD4[+] T cells (Table 1), and one in CD8[+] T cells (Supplementary Table 1a). Based on the known biological significance, three of the mutations (mTOR, NFκB2, and TLR2) were considered as putative driver mutations and potentially important for disease pathogenesis and were studied further.

The previously undescribed somatic missense mutation in mTOR (position 11182160, G to C) changes the amino acid proline 2229 to arginine (Fig. 2a). The variant allele frequency (VAF) was 13.3% among CD4[+] T cells (Table 1). This mutation in exon 48 is located in the kinase domain which has been suggested to be important for signal transduction[14,15]. In addition to mTOR, two other interesting mutations were identified in the NFκB2 and TLR2 genes, although the sequencing coverage of these regions was not optimal (Table 1).

The NFκB2 missense mutation (position 104162075, C to A) leads to a change of the amino acid proline 882 to glutamine. TLR2 missense mutation (position 154625732, G to T) results in a change of the amino acid tryptophan 558 to leucine (Fig. 2a). We also performed exome sequencing of CD4[+], CD8[+] T cells, and NK-cells obtained from the index patient in 2015 (Supplementary Tables 1b–d). Altogether, 17 candidate putative somatic mutations were discovered within the CD4[+] T cell population, including those in the mTOR, TLR2, and NFκB2 genes (Supplementary Table 1).

**Sequencing validation of the somatic mutations**. To further validate the mTOR P2229R, TLR2 W558L, and NFκB2 P882Q mutations, CD4[+] T cells obtained in 2015 underwent standard capillary sequencing. Only the mTOR mutation was confirmed (Fig. 2b) due to the low sensitivity of the assay. Therefore, targeted amplicon sequencing with a coverage up to 100,000X and a sensitivity of 0.5% VAF[16] was applied to all available samples from different time points to establish the dynamics and the lineage specificity of the discovered mutations. The mTOR mutation was detected in CD4[+] T cells that were obtained in 2013, 2015, 2017, and 2019 with VAFs 17.3%, 19.2%, 20.5%, and 23.4%, respectively (Table 2 and Supplementary Table 1). The VAF of the mTOR mutation appeared to increase from 2013 to 2019 regardless of the continuous immunosuppressive therapy

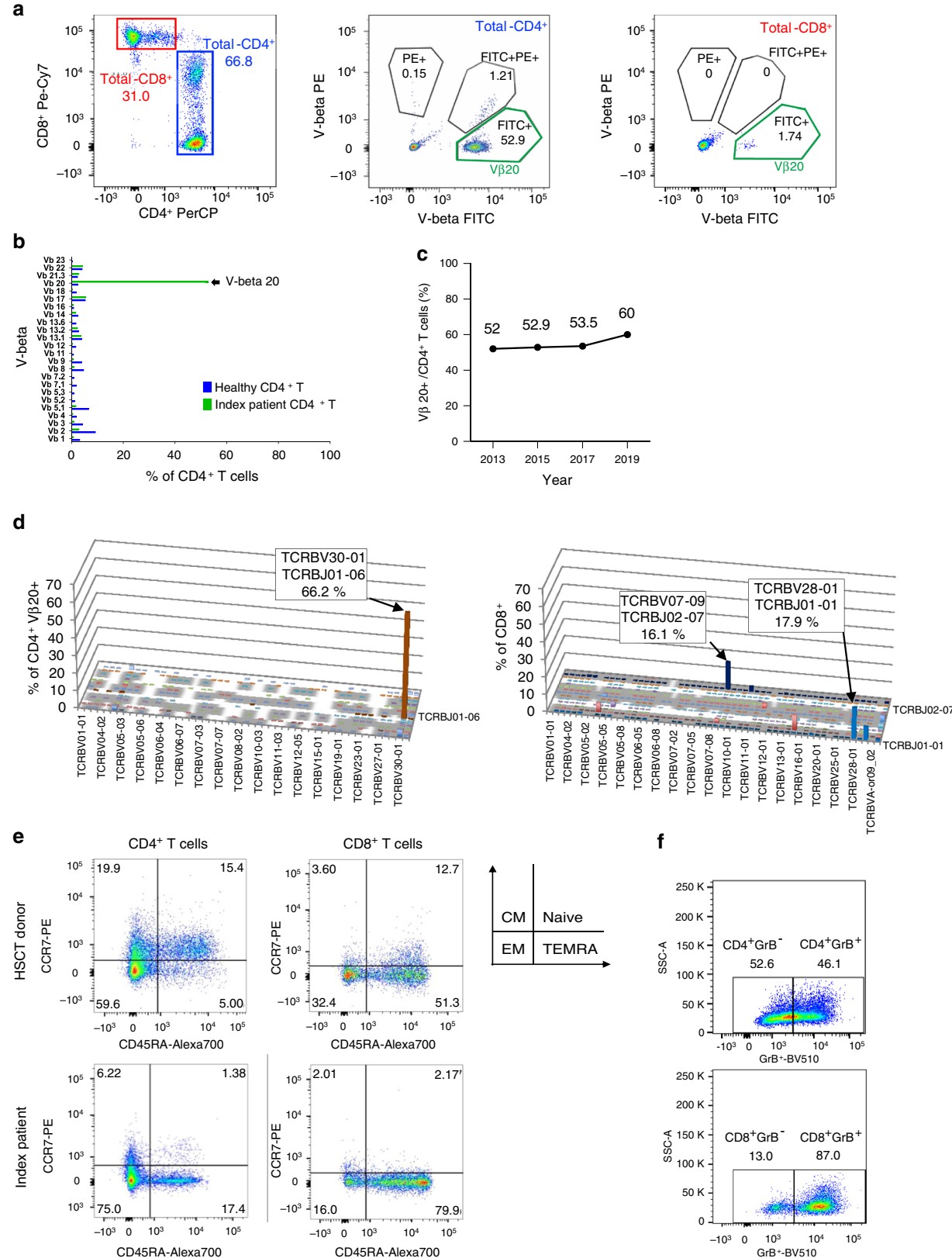

(Fig. 2c). We next verified the *mTOR* mutation by amplicon sequencing from the flow-sorted CD3−, CD8+/CD4−, CD4+Vβ20−, CD4+CD8+ Vβ20−, and monocyte samples. No mutations or very low VAFs were detected (Supplementary Fig. 2b; Supplementary Table 2), suggesting that the mutation was confined to the CD4+ Vβ20+ cell fraction (VAF 44.7% in flow-sorted cells).

Similarly, the *NFκB2* P882Q mutation was confirmed to be limited to CD4+Vβ20+ T cells (VAFs: 12.5% in CD4+ T cells and 21.3% in CD4+Vβ20+ cells) (Table 2 and Supplementary

**Fig. 1 Flow cytometry and TCRβ deep sequencing results from the index patient. a** Gating strategy in screening experiment using the panel of TCR Vβ antibodies. First, CD3 lymphocyte were separated into CD4$^+$ and CD8$^+$ populations using monoclonal antibodies (CD8$^+$: Pe-Cy7, CD4$^+$: PerCP). Next, the samples were stained with Vβ specific antibody cocktails included in IO Test Beta Mark TCR beta Repertoire Kit (Beckman-Coulter Immunotech, USA). The β variable chain family was determined based on FITC and PE positivity from CD4$^+$ and CD8$^+$ populations according to the manufacturer's instruction. Vβ20 clone was detected from total CD4$^+$ T cells (52.9%, middle panel) and total CD8$^+$ T cells (1.74%, right). **b** Flow cytometry Vβ screening results from the index patient's peripheral blood sample. T cell clonality with antibodies which target Vβ region of TCR was analysed of CD4$^+$ T cells. The increased distribution suggests that the cells have large T cell clone. **c** Increased Vβ20 bearing clonotype over time in the index patient's CD4$^+$ T cells. Source data are provided as a Source data file. **d** T cell repertoire of FACS-sorted CD4$^+$Vβ20+ and CD8$^+$ T cells analysed with TCRβ deep sequencing (Adaptive Biotechnologies). The TCRBV30-01 clone was detected in the CD4$^+$Vβ20+ fraction, but not in the CD8$^+$ fraction. **e** Multicolor flow cytometry was applied to identify the immune phenotype of HSCT donor and index patient's memory T cell subtypes. Central memory (CM), naïve, effector memory (EM), and terminal effector memory (TEMRA) cells. **f** The relative proportion of granzyme B positive (GrB$^+$) CD4$^+$ T cells and GrB$^+$CD8$^+$ T cells in index patient. Index patient's PBMCs were stained with anti-CD45, −CD3, −CD4, and −CD8 (surface markers), and then GrB stained after fixation and permeabilization. Stained cells were analyzed using FACSVerse.

Tables 1, 2). In contrast, *TLR2* W558L mutation was discovered in both CD4$^+$ (VAFs: 16.5% in CD4$^+$ cells, 35.5% in CD4$^+$Vβ20 + cells) and CD8$^+$ T cells (VAF 4.7 %) (Supplementary Tables 3, 4). To examine whether the mutations were already present in the HSCT donor, both CD4$^+$ and CD8$^+$ T cells from the donor were sequenced by amplicon sequencing, but no mutations were detected even with very high sequencing coverage (*mTOR, TLR2,* and *NFkB2* total depths: 705,188, 403,633, and 395,669, respectively).

In the course of the disease, the cGvHD affected different organs of the index patient (liver, eyes, and nails), but particularly the skin. To explore whether lymphocytes harboring the detected somatic mutations can be found in target organs, we screened paraffin-embedded biopsy samples by amplicon sequencing. The *mTOR* P2229R mutation was identified in a sclerodermatous skin lesion that was biopsied in 2015 (VAF 0.8% in Fig. 2d; Table 2), but not in eye or liver biopsies. Immunofluorescence staining demonstrated CD4$^+$ and CD8$^+$ T cell infiltration in the same sclerodermatous lesion (Fig. 2e).

**Screening for the somatic mutations in cGVHD patient cohort.** To explore whether the mutations identified in the index patient are recurrent, blood samples from 134 cGvHD patients, 38 allo-HSCT patients without cGvHD, and 54 healthy controls were screened by the amplicon sequencing panel. Two additional cGvHD patients carried the same *mTOR* missense mutation yielding in a *mTOR* P2229R mutation frequency of 2.2% in all cGvHD patients (3 out of 135, including the index patient). In the other two patients, *mTOR* mutation was also detected during active period of cGvHD. Using serial samples of one patient, the detection of the *mTOR* mutation matched the onset of cGvHD, and after the successful corticosteroid therapy the *mTOR* mutation was no longer detectable in the follow-up samples (clinical characteristics described in detail in the Supplementary results). In healthy controls or allo-HSCT patients without cGvHD, no *mTOR* mutations were detected. Furthermore, *NFκB2* mutations were neither detected in additional cGvHD patients nor in healthy controls. The *TLR2* W558L mutation was found in both cGvHD patients (21/134, 15.7%) and healthy controls (5/54, 9.3%), but the VAF indicated a 10-fold and 4-fold higher mutation frequency in cGvHD patients' CD4$^+$ and CD8$^+$ T cells compared to healthy controls (mean 10.5% vs. 1.3% in CD4$^+$ T cells and mean 8.4% vs. 2.8% in CD8$^+$ T cells, Supplementary Fig. 3; Supplementary Tables 3, 4).

**Functional validation of the somatic *mTOR* mutation in vitro.** To examine the functional consequences of *mTOR* P2229R mutant, we transfected HEK293 cells with *mTOR* WT and *mTOR* P2229R mutant constructs. In standard cell culture conditions, cell proliferation was significantly increased in HEK293 expressing *mTOR* P2229R compared to HEK293 expressing *mTOR* WT or empty vector (Fig. 3a, b). Since the kinase activity of mTOR is tightly regulated by nutrient availability, and nutrient starvation leads to downregulation of mTORC1 signaling and induced autophagy[17], we hypothesized that the *mTOR* mutation confers resistance to nutrient starvation-induced cell death. Indeed, HEK293 cells expressing *mTOR* P2229R showed increased cell viability (Fig. 3a, b) together with lower rate of apoptosis (Fig. 3c) upon serum starvation compared to *mTOR* WT expressing HEK293 cells.

As the mTOR signaling pathway consists of two functionally distinct multi-protein complexes, mTOR complex 1 (mTORC1) and mTOR complex 2 (mTORC2)[18], we next investigated whether the *mTOR* P2229R mutant impacts the activation of these complexes and their downstream targets in the HEK293 cell line model. While equal overexpression of wild-type and mutant mTOR was confirmed, no differences in downstream signaling of either complexes were observed in standard culture conditions (Supplementary Fig. 4). In contrast, after 12 h of serum starvation, overexpression of *mTOR* P2229R mutant was associated with activation of both mTORC1 and mTORC2 pathways, as evidenced by increased phosphorylation of mTORC1 targets eukaryotic translation initiation factor 4E (eIF4E)-binding protein 1 (4E-BP1), ribosomal S6 kinase (S6K1) and its downstream target ribosomal protein S6, as well as mTORC2 target AKT (Fig. 3d and Supplementary Fig. 5). We also observed increased phosphorylation and decreased total protein level of Forkhead box protein O1 (FoxO1), a direct target of AKT that regulates T cell differentiation[19]. Tuberous sclerosis complex (TSC) 1 and 2 are negative regulators of mTOR[20], and their levels were slightly decreased in cells expressing the mutant *mTOR* compared to wild-type *mTOR* expressing cells upon serum starvation. The overexpression of *TLR2* (W558L) or *NFκB2* (P882Q) mutants did not influence protein levels of mTOR pathway compared to wild-type proteins, suggesting that the combined effect of triple mutant proteins compared to triple wild-type proteins on mTOR pathway were solely due to mutant *mTOR* (Fig. 3d and Supplementary Fig. 5).

While mTOR is a component of both mTORC1 and mTORC2, the binding partners are different. Regulatory protein associated with mTOR (Raptor) and rapamycin insensitive companion of mTOR (Rictor) bind mTOR in mTORC1 and mTORC2, respectively[18]. No differences in overall protein levels of Raptor or Rictor was observed in HEK293 cells overexpressing *mTOR* P2229R compared to wild-type *mTOR* (Fig. 3d). Furthermore, no differences in mTOR bound Raptor or Rictor was observed after immunoprecipitation between *P2229R* mutant and wild type *mTOR* cells (Fig. 3e). DEP domain-containing mTOR-interacting protein (DEPTOR) is a naturally occurring inhibitor of mTOR

**Table 1 Somatic mutations discovered in CD4+ T cells in the index patient, detected from 2013 sample.**

| Chr | position | Ref | Var | Gene | Mutation type | Codon change | Exon | Amino acid change | CD8_Ref reads[a] | CD8_Alt reads[b] | CD8_Var_Freq (%) | CD4_Ref reads[c] | CD4_Alt reads[d] | CD4_var_Freq (%) | Somatic p-value* |
|---|---|---|---|---|---|---|---|---|---|---|---|---|---|---|---|
| 1 | 1182160 | G | C | MTOR | MISSENSE | cCt/cGt | 48 | P2229R | 179 | 8 | 4.28 | 157 | 24 | 13.26 | 0.0017815 |
| 4 | 154625732 | C | T | TLR2 | MISSENSE | tGg/tTg | 1 | W558L | 116 | 2 | 1.69 | 104 | 10 | 8.77 | 0.014504 |
| 4 | 144801662 | G | G | GYPE | MISSENSE | gGa/gGa | 2 | G13A | 150 | 68 | 31.19 | 147 | 99 | 40.24 | 0.026604 |
| 19 | 50017643 | G | T | FCGRT | MISSENSE | caG/caT | 3 | Q167H | 28 | 0 | 0 | 14 | 2 | 12.5 | 0.12685 |
| 16 | 31388150 | A | G | ITGAX | MISSENSE | Aca/Gca | 21 | T847A | 57 | 0 | 0 | 35 | 2 | 5.41 | 0.15237 |
| 11 | 2415324 | G | T | CD81 | SPLICING | | 4 | | 58 | 0 | 0 | 36 | 2 | 5.26 | 0.15417 |
| 22 | 37326772 | C | A | CSF2RB | MISSENSE | caC/caA | 8 | H310Q | 54 | 0 | 0 | 35 | 2 | 5.41 | 0.16264 |
| 3 | 49936028 | A | C | MST1R | MISSENSE | Tgt/Ggt | 4 | C548G | 45 | 0 | 0 | 38 | 2 | 5 | 0.21849 |
| 10 | 104162075 | C | A | NFKB2 | MISSENSE | cCa/cAa | 23 | P882Q | 14 | 0 | 0 | 26 | 3 | 10.34 | 0.29609 |
| 10 | 18112382 | G | T | MRC1 | MISSENSE | Ggt/Tgt | 2 | G134C | 29 | 0 | 0 | 35 | 2 | 5.41 | 0.31049 |
| 17 | 80274159 | GT | GT | CD7 | FRAME SHIFT | gca/gAca | 3 | A175D | 8 | 0 | 0 | 16 | 3 | 15.79 | 0.33128 |
| 17 | 80274183 | G | C | CD7 | MISSENSE | gCc/gGc | 3 | A167G | 13 | 0 | 0 | 19 | 2 | 9.52 | 0.37433 |
| 12 | 109017698 | TG | A | SELPLG | MISSENSE | aCg/aTg | 2 | T145M | 46 | 2 | 4.17 | 35 | 3 | 7.89 | 0.38939 |
| 17 | 80274161 | T | T | CD7 | FRAME SHIFT | -/- | 3 | | 7 | 0 | 0 | 11 | 2 | 15.38 | 0.41053 |

Immunogene panel sequencing was performed on both CD4+ and CD8+ T cells from the index cGvHD patient. The table shows discovered somatic mutations in the CD4+ cells.
*Chr* chromosome, *ref* reference base, *var* variant base, *freq* frequency
[a]Sequencing reads supporting reference allele in normal sample.
[b]Sequencing reads supporting variant allele in normal sample.
[c]Sequencing reads supporting reference allele in tumor sample.
[d]Sequencing reads supporting variant allele in tumor sample.
* Somatic p-value for somatic/loss of heterozygosity events.

that inhibits both mTORC1 and mTORC2[21]. The level of mTOR bound DEPTOR was decreased in cells expressing *mTOR P2229R* (Fig. 3e), consistent with increased mTOR activity. Taken together, our data in HEK293 cells suggests that the *mTOR* mutant identified in the index patient leads to gain of function of both complexes mTORC1 and mTORC2, conferring increased proliferation in normal culture conditions and reduced apoptosis upon serum starvation (Fig. 3f). These observations are in line with other kinase domain *mTOR* mutations identified in cancer[15].

As the transcription factor *NFκB2* is a critical regulator of inflammation and immune function[22] and toll-like-receptor 2 (*TLR2*) is one of the pattern recognition receptors important for the pathogenesis of many autoimmune diseases and highly expressed in GvHD patients[23], we also wanted to study the functional effects of these mutations. The *NFκB2 P882Q* mutation was located in the c-terminal domain (Fig. 2a), which is known to play an important role in the ubiquitination and partial proteolysis from NFκB2 (p100) to NFκB2 (p52)[24]. In order to determine the molecular balance between these two states, we performed western blotting in HEK293 cells which revealed an increased expression of p52 in cells overexpressing the mutant *NFκB2* compared to wild-type *NFκB2* (Supplementary Fig. 6a, b) indicating a gain-of-function alteration. Similarly, to study the functional consequences of the *TLR2 W558L* mutation, we evaluated alterations in transcriptional regulation by analyzing mRNA expression levels of a subset of *TLR2* downstream targets by qRT-PCR. This demonstrated significantly increased expression levels of *ELK1* and *FOS1* in HEK293 expressing *TLR2 W558L* mutant as compared to *TLR2 WT* (Supplementary Fig. 6c, d).

**Paired scRNA- and TCRαβ sequencing of the CD4+ T cells**. In order to better understand the heterogeneity of the Vβ20 bearing clonotype and the underlying CD4+ T cell compartment in an unbiased manner, we performed simultaneous single-cell RNA and paired TCRαβ sequencing on two time points from 2015 and 2017 for the index patient's CD4+ T lymphocytes from peripheral blood. From the paired sequencing we received 14,111 CD4+ T lymphocytes passing the quality control. To analyze possible differences between the time points, we pooled the samples together and found six distinct CD4+ phenotypes with graph-based clustering. Surprisingly, most of the cells (71.8%) were characterized by cytotoxicity, and lower frequency of naïve cells (23.4%) and regulatory T cells (4.4%) were identified (Fig. 4a, b) concordant with the flow cytometry results (Fig. 1e, f). To address the highly cytotoxic gene expression profile of CD4+ T cells, we also analyzed CD4+ T cells from a healthy donor and could not find similar expression of cytotoxic genes (Fig. 4c, d). The frequency of cells in clusters were stable between the time points indicating resistance to the ongoing immunosuppressive treatment, and only the abundance of cycling cell population showed over two-fold-change between the two time points (Supplementary Fig. 7a). From the TCRαβ-seq we detected TCRαβ, TCRα or TCRβ in 3577 unique T cell clonotypes. The clonotype matching to TCRβ sequencing data from Vβ20-sorted population (Fig. 1d) harboring the *mTOR*-mutation was also the most expanded fraction, representing 40.1% of the total CD4+ T cells from two time points. Almost all of the cells from this clonotype (98.4%) belonged to the cytotoxic clusters, and most of the cells were included in the cytotoxic effector cluster (85.8%) (Fig. 4e).

To understand the effect of the *mTOR*-mutation on the T cells, we performed differential expression (DE) analysis between the cytotoxic cells, comparing the cells from clonotype of interest against the other cytotoxic cells in cytotoxic clusters (effector, effector/effector memory, and memory). The analysis found

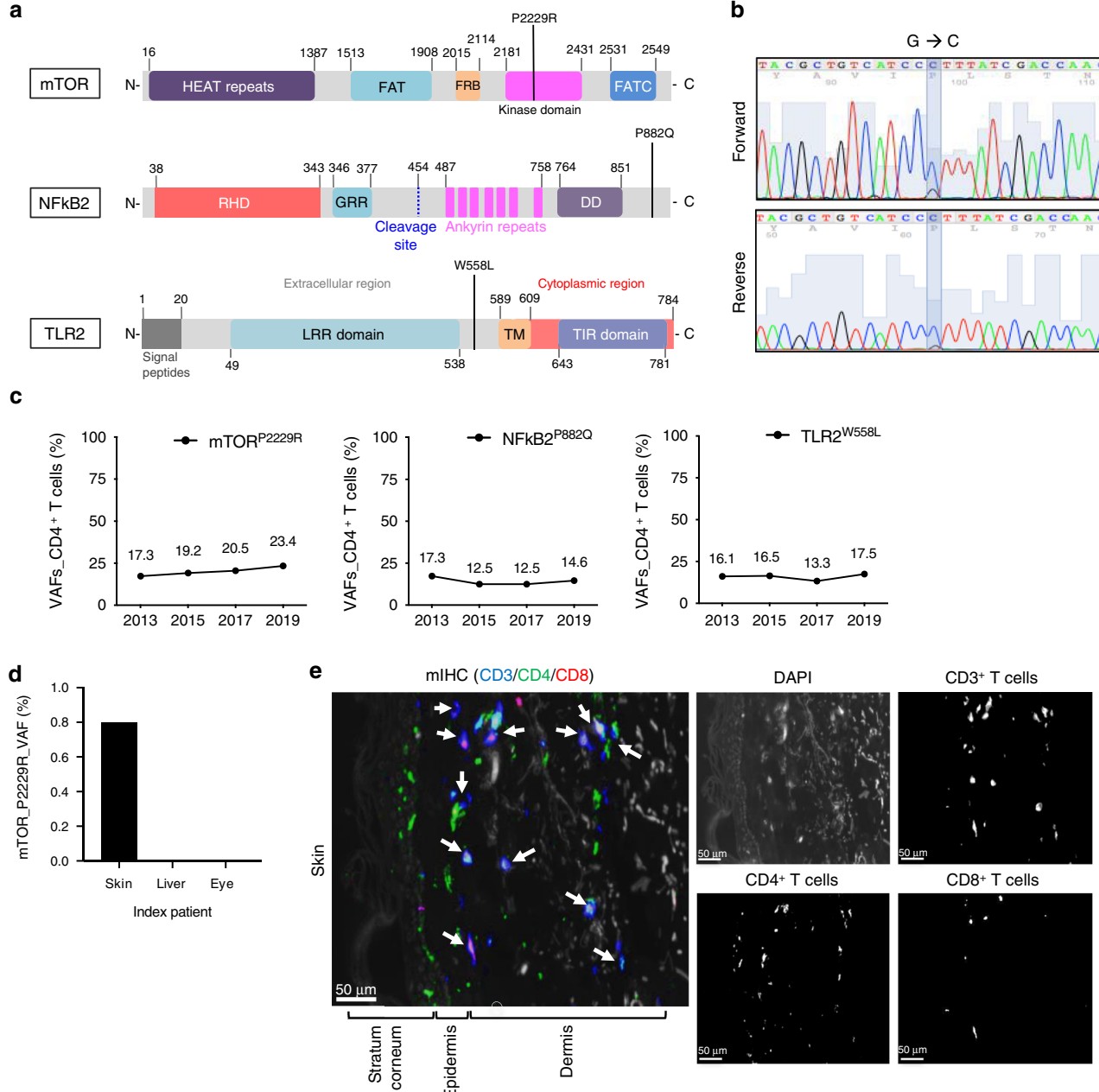

**Fig. 2 mTOR, NFκB2, and TLR2 mutations in index patient. a** Locations of *mTOR*, *TLR2*, and *NFκB2* somatic mutations. Linearized structure of *MTOR*, *NFκB2*, and *TLR2* presenting the location of somatic mutations. *mTOR P2229R* mutation is located in the kinase domain, *NFκB2 P882Q* in the C-terminus, and *TLR2 W558L* between LRR (Leucine-rich repeats) domain and transmembrane (TM) domain. **b** A heterozygous *mTOR* mutation (G to C, *P2229R*) was detected in CD4+ T cells by Sanger sequencing. **c** Variant allele frequencies (VAFs) of *mTOR P2229R*, *NFκB2 P882Q*, and *TLR2 W558L* mutations in the index patient's CD4+ T cells over time as measured with amplicon sequencing. Source data are provided as a Source data file. **d** VAFs (%) of *mTOR P2229R* mutation from the index patient's skin, liver, and eyes biopsy. **e** Immunofluorescence staining indicated CD3+CD4+ and CD3+CD8+ T cell infiltration in the skin. Paraffin embedded skin biopsy from the index patient was sectioned and stained with antibody specific human CD3 (cyan), CD4 (green), and CD8 (red). White arrows indicate infiltrated CD3+CD4+ or CD3+CD8+ T cells. Original imaging magnification: 20×, the figure has been further zoomed to 25× for visualization. Scale bar: 50 μm.

1190 statistically significant DE-genes, of which 601 were upregulated in the clonotype, including cytotoxic genes (e.g., *GZMA*, *GZMB*, *GNLY*, *GZMK*, *NKG7*, and *PRF1*) (Fig. 4f). Additionally, upregulated eukaryote elongation factors (*eEFs*), such as *EEF1A1*, *EEF1B2*, and *EEF2*, supported abnormal growth and proliferation of the expanded CD4+ T cells. Furthermore, the expression of *DUSP2* and *KLRB1* genes was highly specific for the mutated clone (Fig. 4b). To identify differential pathway regulation in the clonotype, Gene Set Enrichment Analysis

(GSEA) was performed, resulting in 11 significantly over-represented and 0 under-represented pathways in the clonotype (Supplementary Table 5). The upregulated pathways included *MYC* target pathways, *TNFα* signaling via *NFκB*, *IL2-STAT5* signaling and hypoxia, supporting that the Vβ20 bearing clonotype is clearly distinct from the other cytotoxic cells (Fig. 4g).

**Cytotoxic GrB+CD4+ T cells are common in patients with cGvHD.** As we observed cytotoxic gene expression signature in

**Table 2 Somatic *MTOR* and *NFκB2* mutations validated by amplicon sequencing in the index patient.**

| Sample year | DNAs | Gene | Chr | Position | ref | var | Amino acid change | Call_Depth | Ref_Calls | Var_Calls | VAF (%) | Freq_Ratio |
|---|---|---|---|---|---|---|---|---|---|---|---|---|
| 2013 | CD4+ T | mTOR | 1 | 112,182,160 | G | C | P2229R | 7204 | 5939 | 1265 | 17.3 | 0.99757 |
| 2013 | CD4+ T | NFκB2 | 10 | 104,162,075 | C | A | P882Q | 52 | 43 | 9 | 17.3 | 0.67697 |
| 2015 | CD4+ T | mTOR | 1 | 112,182,160 | G | C | P2229R | 1,000,004 | 8,08,220 | 191784 | 19.2 | 1.00047 |
| 2015 | CD8+ T | mTOR | 1 | 112,182,160 | G | C | P2229R | 1,000,003 | 9,42,683 | 57320 | 5.7 | 0.9993 |
| 2015 | Skin-biopsy | mTOR | 1 | 112,182,160 | G | C | P2229R | 8085 | 7986 | 99 | 0.8 | 0.96305 |
| 2015 | Whole blood | mTOR | 1 | 112,182,160 | G | C | P2229R | 719 | 703 | 16 | 2.0 | 1.02671 |
| 2015 | CD4+ T | NFκB2 | 10 | 104,162,075 | C | A | P882Q | 4,71,375 | 4,14,821 | 56554 | 12.5 | 1.0061 |
| 2015 | CD8+ T | NFκB2 | 10 | 104,162,075 | C | A | P882Q | 5,86,903 | 5,77,283 | 9620 | 1.6 | 1.00124 |
| 2016 | Whole blood | mTOR | 1 | 112,182,160 | G | C | P2229R | 2743 | 2715 | 28 | 1.0 | 1.01793 |
| 2017 | CD4+ T | mTOR | 1 | 112,182,160 | G | C | P2229R | 7330 | 5796 | 1534 | 20.5 | 0.9141 |
| 2017 | CD4+Vb.20+ | mTOR | 1 | 112,182,160 | G | C | P2229R | 1,68,592 | 93,183 | 75409 | 44.7 | 0.99928 |
| 2017 | CD4+CD8 +Vb.20+ | mTOR | 1 | 112,182,160 | G | C | P2229R | 1,71,728 | 1,10,670 | 61058 | 35.5 | 0.99910 |
| 2017 | CD4+ T | NFκB2 | 10 | 104,162,075 | C | A | P882Q | 1436 | 1236 | 200 | 12.5 | 0.82101 |
| 2017 | CD4+Vb.20+ | NFκB2 | 10 | 104,162,075 | C | A | P882Q | 26,262 | 20,668 | 5594 | 21.3 | 1.00395 |
| 2017 | CD4+CD8 +Vb.20+ | NFκB2 | 10 | 104,162,075 | C | A | P882Q | 48,654 | 43,822 | 4832 | 9.9 | 1.00282 |
| 2019 | CD4+ T | mTOR | 1 | 112,182,160 | G | C | P2229R | 6890 | 5261 | 1629 | 23.4 | 0.99281 |
| 2019 | CD4+ T | NFκB2 | 10 | 104,162,075 | C | A | P882Q | 82 | 70 | 12 | 14.6 | 1.00724 |

CD4+ and CD8+ T cells were sorted either with the magnetic beads or flow-based sorting (2017 sample). In addition, CD4+Vb20+ and CD4+CD8+Vb20+ fractions were sorted with flow cytometry (2017 sample). *MTOR* and *NFκB2* mutations were analysed from sorted fractions with deep amplicon sequencing. Mutations were confined to CD4+ fractions. The low mutation VAFs in CD8+ fractions are due to small CD4+ T cell contamination (CD4+CD8+ double positive cells) in the bead sorted fraction.
*Chr* chromosome, *ref* reference base, *var* variant base, *Call_depth* total number of called reads, *Ref_calls* sequencing reads supporting reference allele, *Var_calls* sequencing reads supporting variant allele, *VAF* variant allele frequency, *Freq_frequency*, *Freq_Ratio* base quality frequency ratio

index patient's CD4+ T cells, we wanted to confirm this finding by GrB flow cytometry staining in other cGvHD patients' CD4+ and CD8+ T cells. Interestingly, the GrB positivity was high both among cGvHD patients' CD4+ (median 46.1%) and CD8+ T cells (median 87.0%) when compared to healthy controls (median 2.0% and median 12.8%, respectively, Supplementary Fig. 8a, b). We also studied the cytokine production capability of GrB+ cells and noted that patients' cells were highly active in secreting Th1-type cytokines (TNF-α and IFN-γ)(Supplementary Fig. 8c, d). Index patient's CD4+GrB+ T cells were among the top cytokine producers when compared to other cGvHD patients (Supplementary Fig. 8c).

**CD4+ T cells are cytotoxic against primary fibroblasts.** Real-time electrical impedance measurements monitoring target cell killing have been widely applied to study the cellular cytotoxicity in vitro[25,26]. To test the functional effects of the mutated CD4+ T cells, we performed co-culture experiments with CD4+ and CD8+ T cells and primary fibroblasts from the index patient. Addition of purified CD4+ T cells on the monolayer of primary fibroblasts resulted in dose-dependent decrease in electrical impedance, implicating cytotoxicity of the CD4+ T cells (Fig. 5a and Supplementary Fig. 9), a recently defined phenomenon[27]. In contrast, the CD8+ T cells showed no cytotoxic activity against the fibroblasts, as the impedance curve mirrored the control well without effector cells (Fig. 5a and Supplementary Fig. 9a). To confirm whether CD4+ T cell cytotoxicity was specific to the index patient, we performed the cytotoxicity analysis for three healthy donors. CD4+ and CD8+ T cells from healthy donors did not show cytotoxic activity against fibroblasts extracted from the same healthy individuals (Fig. 5c and Supplementary Fig. 9b–d).

We next investigated whether the cytotoxicity of the index patient's CD4+ T cells was mediated through major histocompatibility complex (MHC) class I or MHC class II by culturing the patient fibroblasts with HLA I (HLA-ABC) specific or HLA II (HLA-DR, DP, and DQ) specific monoclonal antibodies before the addition of CD4+ T cells. HLA I monoclonal antibody partially rescued the fibroblast viability whereas no impact was observed with HLA II specific antibody (Fig. 5d). While the exact mechanism of this effect remains to be studied in more detail, these data suggest MHC class I mediated fibroblast killing by cytotoxic CD4+ T cells in the index patient.

**Drug sensitivity and resistance testing in CD4+ T cells.** To determine sensitivity of the mutated cells to targeted therapy, we performed robust ex vivo drug sensitivity and resistance testing (DSRT) with 527 drugs in five different concentrations[28]. Freshly isolated CD4+ T cells from the index patient, the HSCT donor, and a healthy control were used. In this screen, the index patient's CD4+ T cells were less sensitive to mTOR/PI3K inhibitors as compared to the HSCT sibling donor's CD4+ T cells (Fig. 6a, b), although constitutive PI3K/AKT/mTOR activity generally predicts rapalog sensitivity. Instead, heat shock protein 90 (HSP90) inhibitors showed an increased killing effect on the index patient's CD4+ T cells as compared to CD4+ T cells from both the HSCT donor and healthy control (Fig. 6a, c, d). With regard to other clinically interesting drug classes, both the HSCT donor and the recipient CD4+ T cells were sensitive to HDAC inhibitors, CDK-inhibitors, tyrosine kinase inhibitor, and proteosome inhibitors. The CD4+ T cells from HSCT donor were also modestly more sensitive to glucocorticoids (dexamethasone and methylprednisone), but this was not statistically significant. Neither HSCT donor nor recipient CD4+ T cells were sensitive to cyclophosphamide, tacrolimus, or methotrexate. However, it should be considered that the assay read-out is cell death, and

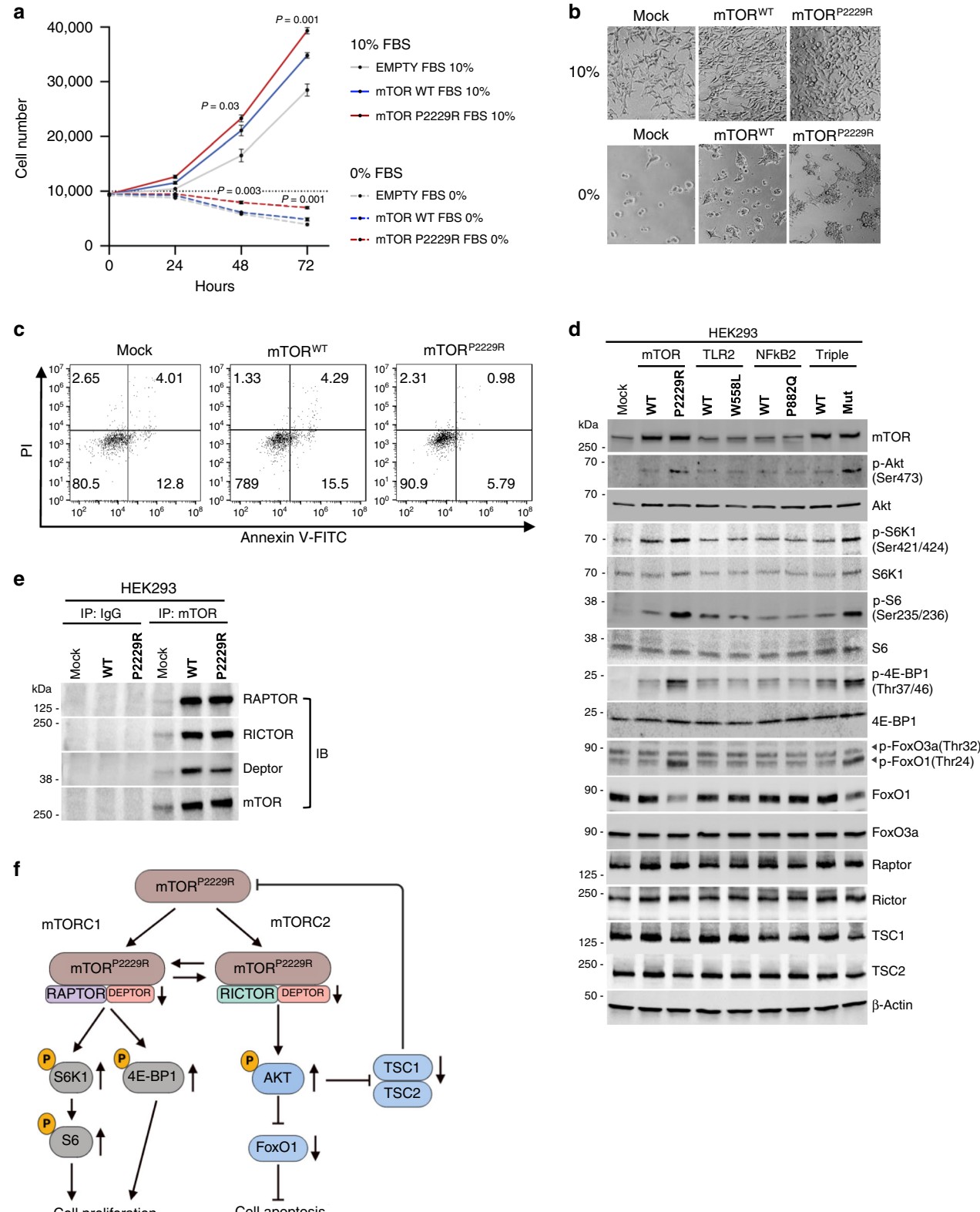

lymphocytes were not activated nor actively proliferating during the experiment. To elaborate these results, we investigated whether mTOR or HSP90 inhibitors reduced the cytotoxicity of CD4$^+$ T cells. For this purpose, we performed the real-time cytotoxicity assay with the index patient's fibroblasts and CD4$^+$ T cells, with drugs added simultaneously with CD4$^+$ T cells to reduce direct effects on fibroblasts. Both ganetespib (100 nM) and

sirolimus (100 nM) attenuated CD4$^+$ T cell cytotoxicity (Fig. 6e and supplementary Fig. 10). In line with DSRT result (Fig. 6a–d), the inhibition of fibroblast killing was slightly higher with ganetespib compared to sirolimus.

Next, we studied the effects of these candidate drugs on mTOR pathway in HEK293 cells. In cells stably expressing *mTOR WT* and *mTOR P2229R* mutant, ganetespib (HSP90

**Fig. 3 Functional analysis of *mTOR*, *TLR2*, and *NFκB2* mutants in HEK293 cells. a** HEK293 cells stably expressing empty vector (Mock), *mTOR WT*, and *P2229R* ($1 \times 10^4$ cells) were incubated with or without serum for indicated times. Cell proliferation was measured by DNA fluorescence-based assay. Cell numbers were calculated with a standard curve from the measured fluorescence intensity. Error bar present mean ± SD of three independent experiments. *p* values were calculated using unpaired *t*-test (*mTOR*[P2229R] vs. *mTOR*[WT]). Source data are provided as a Source data file. **b** Cell morphology after 72 h. **c** FACS analysis by Annevin V-FICT/PI double staining of HEK293 cells (Mock, *mTOR*[WT], and *mTOR*[P2229R]) following serum-starvation for 72 h by BD Accuri™ C6 Plus. Early (Annexin V+/PI−) and late apoptosis (Annexin V+/PI+) are indicated as percentages. **d** Stably expressed *mTOR*[WT], *mTOR*[P2229R], *TLR2*[WT], *TLR2*[W559L], *NFκB2*[WT], *NFκB2*[P882Q], *Triple*[WT] (*mTOR*[WT], *TLR2*[WT], and *NFκB2*[WT]), and *Triple*[MUT] (*mTOR*[P2229R], *TLR2*[W559L], and *NFκB2*[P882Q]) in HEK293 cells were serum starved for 12 h. The cell lysates were subjected to western blot analysis using the indicated antibodies. Data is representative of three independent experiments. Summary bar plots of the experiment are shown in Suppl. Fig. 5. Source data are provided as a Source data file. **e** Co-Immunoprecipitation (Co-IP) of HEK293 stably expressing *mTOR WT* and *P2229R*. HEK293 cells stably expressing *mTOR*[WT] and *mTOR*[P2229R] are serum starved for 12 h, and mTOR was immunoprecipitated (IP) with anti-mTOR antibody (1:50 dilution) and immunoblotted (IB) for RAPTOR, PIRCTOR, and DEPTOR. Respective plots without starvation are shown in Suppl. Fig. 4. Source data are provided as a Source Data file. **f** Simplified scheme of *mTOR P2229R* mutation in the mTOR pathway. Pathway components discovered to be upregulated or downregulated with functional assays are marked with arrows. Akt protein kinase B, TSC tuberous sclerosis complex, mTOR mammalian target of rapamycin, mTORC1 mTOR complex 1, mTORC2 mTOR complex 2, Raptor regulatory associated protein of mTOR; rapamycin insensitive companion of mTOR, DEPTOR DEP domain containing mTOR interacting protein, S6K1 p70 ribosomal S6 kinase 1, S6 S6 ribosomal protein, 4E-BP1 eIF4E-binding protein 1, FoxO1 forkhead family of transcription factor 1.

inhibitor) reduced AKT phosphorylation on serine 473. AKT phosphorylation appeared to be normal following sirolimus treatment as rapalogs only inhibit mTORC1, but not mTORC2[29] activity. Treatment with either drug resulted in decreased levels of phosphorylated S6K1 in both mutant and WT cells (Fig. 6f). Both drugs also led to a reduced pS6 phosphorylation in CD4[+] T cells from the index patient and a healthy control (Fig. 6g), suggesting an inhibitory effect on mTORC1 activity. AKT was more phosphorylated in the patient's CD4[+] T cells as compared to controls with a slight decrease in both samples upon treatment with ganetespib or sirolimus.

## Discussion

Chronic GvHD remains a major clinical challenge after allo-HSCT. Detailed characterization of the immune system using new analysis tools can give insights into the pathophysiology of the disease. Peripheral expansion of cytotoxic T cells from the donor graft is a previously described phenomenon[30]; however, whether this clonal expansion is associated with genetic evolution has not been comprehensively studied.

Here we coupled next-generation DNA sequencing, single-cell RNA sequencing and TCR sequencing to carefully characterize the immune phenotype of an index patient with cGvHD. We identified three somatic candidate mutations in clonally expanded CD4[+] T cells: *mTOR, TLR2*, and *NFκB2*. The *mTOR* mutation was recurrent and found in total in 2.2% of cGvHD patients. In the index patient, the mutated clone expanded during the course of active cGvHD despite immunosuppressive treatment and was present in both blood and sclerodermatous skin lesion samples suggesting its contribution to disease pathogenesis. No mutations were discovered in the sibling donor samples proposing the mutations been formed after the allo-HSCT. Functional in vitro studies indicate that the *mTOR* mutation results in a gain-of-function alteration activating both mTORC1 and mTORC2 pathways.

Accumulation of somatic mutations is inherently associated with normal cell division. The role of somatic mutations in cancer is well established, and recent reports suggest that somatic mutations may also play a role in the pathogenesis of non-malignant diseases[31–35]. It was also recently shown that the disruption of the *TET2* gene by lentiviral vector-mediated insertion of the chimeric antigen receptor (CAR) transgene led to the expansion of single CAR T cell in a patient with chronic lymphocyte leukemia and enhanced therapeutic efficacy[36].

Given the role of mTOR in T cell homeostasis and integrating immune signals and metabolic clues[37], we further characterized the novel *mTOR P2229R* mutation in detail. The P2229R

mutation was located in the kinase domain where cancer-associated gain-of-function mutations have been shown to lead to activation of both mTORC1 and mTORC2 pathways[15,38,39]. Consistent with these prior reports, overexpression of *mTOR P2229R* in HEK293 cells resulted in increased cell proliferation and cell survival compared to wild type *mTOR*. The mutant protein effectively activated both mTORC1 and mTORC2 downstream signaling pathways in HEK293 cells upon serum starvation whereas no differences in the amount of mTOR bound Raptor or Rictor were observed, suggesting that the mutation did not influence complex formation. In contrast, the amount of mTOR bound Deptor, a naturally occurring mTOR inhibitor, was decreased in *mTOR P2229R* mutant cells, consistent with gain-of-function nature of the mutation[15].

The other two genes mutated in clonal T cells of the index patient are also implicated in inflammation[40,41]. The index patient harbored *NFκB2* (p100) *P882Q* somatic mutation that is located in the C-terminus, and our HEK293 model with over-expression of the mutant *NFκB2* suggested increased p52 formation, potentially inducing hyperactivation of non-canonical NF-κB pathway. Unlike the discovered *mTOR* and *NFκB2* mutations, the *TLR2 W558L* somatic mutation was identified in both cGvHD patients and healthy controls; however, VAFs were significantly higher in cGvHD patients as in healthy controls (Supplementary Tables 3, 4). While further studies are warranted to elucidate the relevance of these mutations in cGvHD pathogenesis, their discovery highlights clonal evolution in the expanded T cell clone.

Paired scRNA-seq and TCRαβ-seq of the index patient's blood samples showed that the majority of the expanded CD4[+] T cells had upregulated expression of genes associated with cytotoxicity and cellular proliferation. This was further supported by the ex vivo cytotoxicity assay where the index patient's mutated CD4[+] T cells, but not CD8[+] T cells, possessed dose-dependent cytotoxicity against the patient's own primary fibroblasts. Further experimentation showed partial rescuing of the cytotoxic phenotype when fibroblasts where cultured with MHC class I, but not class II, antibodies before T cells were added. Flow cytometry staining confirmed that GrB[+] CD4[+] putative cytotoxic T cells are common in patients with cGvHD. While the exact mechanism of this phenomenon is unclear, these data suggest a role for recently discovered cytotoxic CD4[+] T cells[42] in GvHD pathogenesis.

The index patient continued to suffer from cGvHD manifestations despite multiple lines of immunosuppressive therapy, including mycophenolate, prednisone, and cyclosporine. Using primary patient CD4[+] T cells, we performed a large-scale drug sensitivity screen to identify potential targeted therapies.

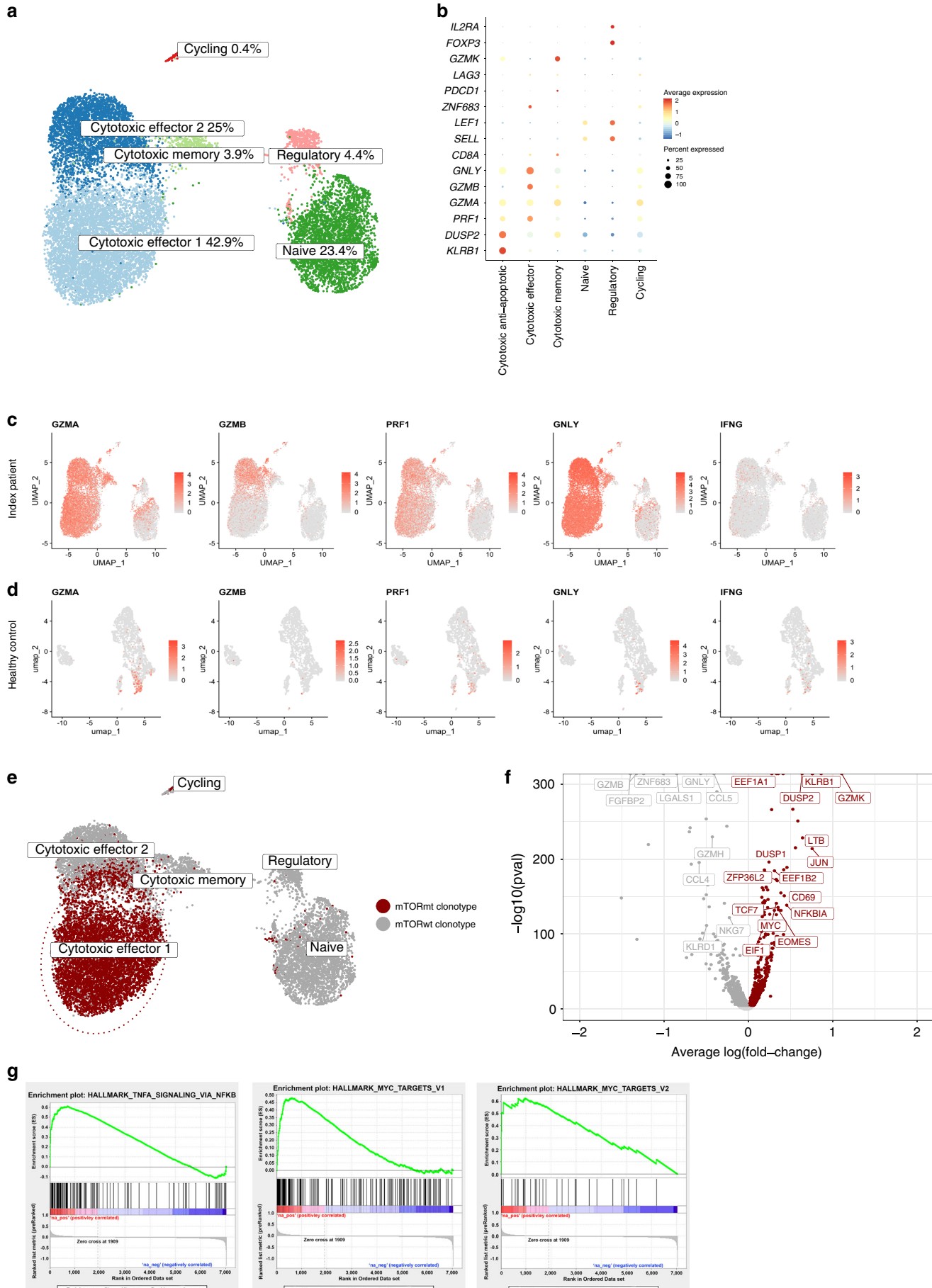

**Fig. 4 Single-cell RNAseq analysis from the index patient. a** Two-dimension UMAP-projection of clustered CD4+ cells pooled from two time points from peripheral blood. A total of 14,111 cells are annotated in six distinct clusters, three of which can be annotated as cytotoxic based on expression of genes such as *GZMA, GZMB,* and *GNLY*. The read counts of scRNA seq data are provided as source file. **b** Gene expression heatmap for the six distinct CD4+ clusters, where rows represent canonical marker genes and columns represent different clusters. **c** Feature plot showing the scaled expression of cytotoxic marker genes across 14,111 CD4+ T cells from the index cGvHD patient in the UMAP embedding. **d** Feature plot showing the scaled expression of cytotoxic marker genes across 2322 CD4+ T cells from a healthy donor in the UMAP embedding. **e** Graphical visualization showing the cells taken into differential expression analysis. Red shows the Vβ20 bearing, *mTOR* mutated clonotype, and gray cells represent the cells from other clonotypes with similar, cytotoxic phenotype as the cells from the mutated clonotype. **f** Volcano plot showing differentially expressed genes between clonotype of interest (red) and cells from other clonotypes with similar, cytotoxic phenotype as the cells from the mutated clonotype (gray). **g** Gene Set Enrichment Analysis (GSEA) results from the differential expression analysis. Shown here are three of six HALLMARK-categories enriched (FDR *q*-val < 0.05) Vβ20 bearing, *mTOR* mutated clonotype.

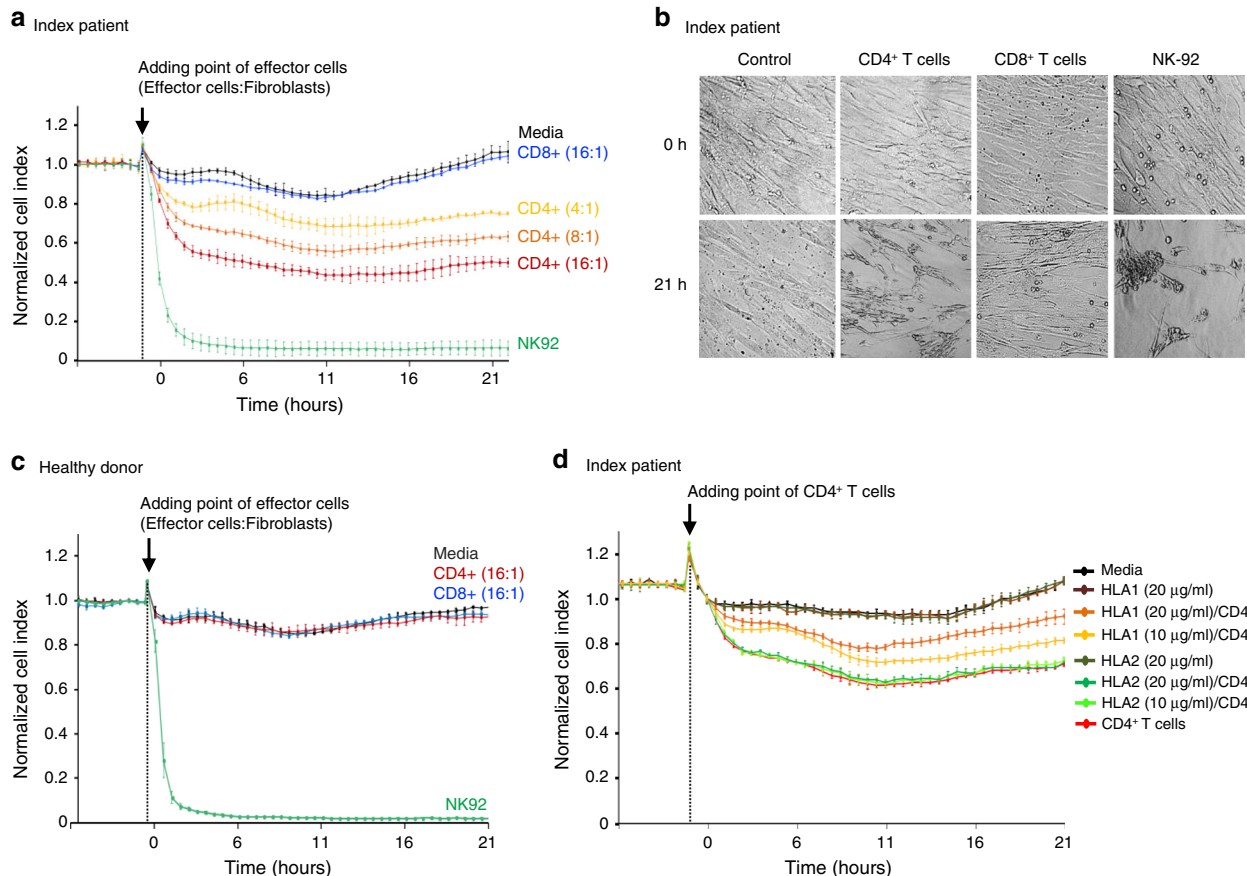

**Fig. 5 Real-time monitoring of cellular cytotoxicity by electrical impedance measurement.** Real-time cell analysing (RTCA) systems, xCELLigence™, was applied to monitor real-time killing effect of primary fibroblasts obtained from the index patient and healthy donors. **a** Index patient's primary fibroblasts were cultured as monolayers for 24 h to reach full confluence. Once confluent, the effector cells; NK-92 cell (positive control, 8:1), primary CD4+ T cells (4, 8, 16:1) and primary CD8+ T cells (16:1) with different ratios (effector cells:fibroblasts) were added to each well followed by co-culture (arrow indicates the point of adding the effector cells, 0 h). The control (black line, media) shows the impedance of the fibroblasts without any added effectors. The cell impedance was measured every 30 min for 21 h. The measured impedance was expressed as Cell Index with the normalization performed at time of addition of effector cells. **b** Visualization of monolayers of the primary fibroblast before and 21 h after addition of effector cells (8:1 for all effector cells). **c** CD4+ T cells and CD8+ T cells from healthy donors were added to healthy donors' own fibroblasts (*n* = 3). The cell impedance was measured for 21 h. Data is representative of three independent individuals. **d** Index patient's fibroblasts were seeded with HLA1 and HLA2 antibodies (10 and 20 µg/mL). After CD4+ T cells (CD4+ T cells:fibroblasts = 8:1) were added the cell impedance was measured for 21 h. Dots represent mean values and error bars indicate range (*n* = 2 for all conditions, technical duplicates). Source data are provided as a Source data file.

Compared to HSCT donor and healthy control, the index patient's CD4+ T cells were more sensitive to HSP90 inhibitors. Consistent with these results, HSP90 inhibitor ganetespib conferred lower killing of primary fibroblasts by CD4+ T cells in the cytotoxicity assay. HSP90 inhibition has shown efficacy in a mouse model of GvHD[43], and although our results are from a single patient, our data support the role for HSP90 inhibition in treating cGvHD. Of note, we also observed resistance of patient

CD4+ T cells to mTOR/PI3K inhibitors compared to donor cells, consistent with hyperactivation of mTOR pathway due to the *P2229R* mutation.

While somatic mutations in clonally expanded T cells in transplant recipients suffering from cGvHD is an interesting observation, the generalizability of this phenomenon in cGvHD remains to be established by further studies. *mTOR* was mutated in 2.2% (3 out of 135) of cGvHD patients, indicating recurrence of

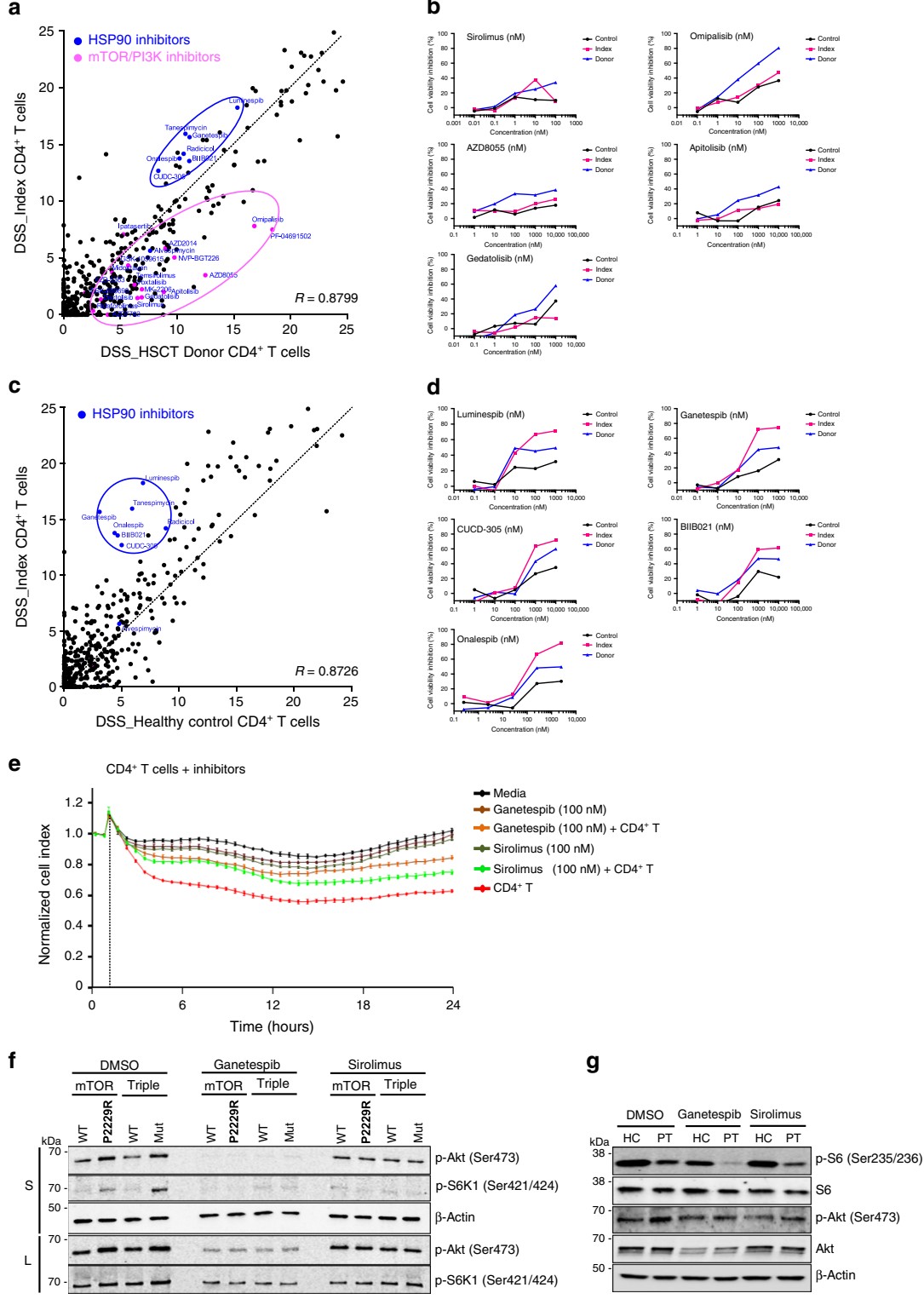

these mutations. Furthermore, these *mTOR* mutations were detected during active cGvHD emphasizing their role in the pathogenesis. Whether somatic mutations in cytotoxic T cells are associated with cGvHD severity, drug responses, and clinical outcome will have to be validated in larger cGvHD cohorts. Sequential samples starting from graft are also needed to understand the time of mutation formation and clonal evolution in detail.

In conclusion, *mTOR, NFκB2*, and *TLR2* somatic mutations were discovered in an expanded CD4+ T cell clone in a patient

with active cGvHD. *mTOR* mutation was confirmed to be recurrent is a small subset of other cGvHD patients, whereas it was not observed in healthy donors or allo-HSCT patients without cGvHD. Clonal somatic mutations may also exist in other genes and cell subsets, warranting further studies to elucidate the extend of this phenomenon in the setting of alloimmunity. Our findings imply somatic mutations as one putative mechanism for the aberrant, persistent T cell activation in cGvHD and pave the way for potential individualized therapies.

**Fig. 6 Drug Sensitivity and Resistance testing (DSRT) of CD4+ T cells from index patient compared with CD4+ T cells from both healthy control and HSCT donor.** Ex vivo DSRT was performed on fresh CD4+ T cells from index patient, HSCT donor, and healthy control. Correlation of drug sensitivity scores (DSS) indicating cell viability inhibition measured by CellTiter-Glo 2.0 (Promega, USA). DSS is a quantitative measurement of a drug response based on the area under the curve (AUC) with further normalization. Higher DSS denote better killing activity. **a** Correlation of DSS scores between index patient and HSCT donor CD4+ T cells. **b** Individual dose response curves of index patient, HSCT donor, and healthy control CD4+ T cells for mTOR inhibitors. **c** Correlation of DSS scores between index patient and healthy control CD4+ T cells. **d** Individual dose response curves of index patient, HSCT donor, and healthy control CD4+ T cells for HSP90 inhibitors. **e** Ganetespib (100 nM) and sirolimus (100 nM) were added with CD4+ T cells after 24 h when the index patient's fibroblasts reached full confluence. The cell impedance was measured for 24 h. Dots represent mean values and error bars indicate range (n = 2 for all conditions, technical duplicates). CD4+ T cells:fibroblasts = 8:1. **f** Stably expressed $mTOR^{WT}$, $mTOR^{P2229R}$, $Triple^{WT}$ ($mTOR^{WT}$, $TLR2^{WT}$, and $NFKB2^{WT}$) and $Triple^{MUT}$ ($mTOR^{P2229R}$, $TLR2^{W559L}$, and $NFKB2^{P882Q}$) in HEK293 cells were treated with HSP90 inhibitor (Ganetespib, 100 nM) or mTOR inhibitor (Sirolimus, 100 nM) for 12 h. Western blot was performed with the use of anti-pAkt, anti-pS6K1, and anti-β-actin antibodies. Different amount of total protein was loaded in the upper panel (S, 30 μg) and the lower panel (L, 50 μg). Data is representative of three independent experiments. **g** Isolated CD4+ T cells from healthy control (HC) and index patient (PT) were treated with HSP90 inhibitor (Ganetespib, 100 nM) or mTOR inhibitor (Sirolimus, 100 nM) for 12 h. Cells were lysed and proteins were run on the SDS-PAGE gel. Western blot was performed with the use of anti-S6, anti-pS6, anti-Akt, anti-pAkt, and anti-β-actin antibodies. Source data are provided as a Source data file.

## Methods

**Study patients**. Index patient's clinical characteristics and medical history are described in detail in Supplementary results and Supplementary Fig. 1. The clinical characteristics of the other two *mTOR* mutated patients are also described in detail in Supplementary results. Blood samples from 135 patients who had developed cGvHD after allo-HSCT were collected between 2007 and 2016 (Helsinki University Hospital, Helsinki, Finland, n = 8; Turku University Hospital, Turku, Finland, n = 37; Hospital de la Princesa, Madrid, Spain, n = 19; Hospital Morales Meseguer, Murcia, Spain, n = 71). In addition, 38 patients who had not developed cGvHD after allo-HSCT until the date of sampling, served as a control cohort (Turku University Hospital n = 6 and Hospital Morales Meseguer n = 32). The blood samples were collected 3–102 months (mean 13.5 months) and 2–47 months (mean 14.5 months) after allo-HSCT for GvHD and non-cGvHD patients, respectively. Clinical characteristics of the patients are summarized in Supplementary Table 6. All patients provided written informed consents. Additionally, buffy coat samples from 54 healthy blood donors were obtained from the Finnish Red Cross Blood Service. A peripheral blood sample from the index patient's sibling donor (HSCT donor) was also obtained.

The study was performed in compliance with the principles of Helsinki declaration, and was approved by the ethics committees in the Helsinki University Hospital (Helsinki, Finland), Turku University Hospital (Turku, Finland), Hospital de la Princesa (Madrid, Spain), and Hospital Morales Meseguer (Murcia, Spain).

**Antibodies**. Primary antibodies against NFκB2 (Cat#: 4882S, Lot: 4), ribosomal protein S6 (Clone: 54D2, Cat#: 2317S, Lot#: 4), phospho-S6 ribosomal protein (Ser235/236) (Clone: D57.2.2E, Cat#: 4858T, Lot#: 16), Akt (Clone: C67E7, Cat#: 4691T, Lot#: 20), phospho-Akt (Ser473) (Cat#: 9271T, Lot#: 14), p70S6 kinase (S6K) (Clone: 49D7, Cat#: 2708T, Lot#: 7), phospho-p70S6 (S6K) kinase (Thr421/Ser424) (Cat#: 9204S, Lot#: 11), mTOR (Clone: 7C10, Cat#: 2983, Lot#: 16), p-4E-BP1 (Clone: 236B4, Cat#: 2855, Lot#: 26), 4E-BP1 (Cat#: 9452, Lot#: 12), Rictor (Clone: 53A2, Cat#: 2114, Lot#: 7), Raptor (Clone: 24C12, Cat#: 2280, Lot#: 13), phospho-FoxO1 (Thr24)/FoxO3a (Thr32) (Cat#: 9464, Lot#: 7), FoxO1 (Clone: C29H4, Cat#: 2880, Lot#: 11), FoxO3a (Clone: 75D8, Cat#: 2497, Lot#: 8), TSC1 (Clone: D43E2, Cat#: 6935, Lot#: 3), TSC2 (Clone: D93F12, Cat#: 4308, Lot#: 6), and Rabbit IgG Isotype control (Clone: DA1E, Cat#: 3900, Lot#: 34) were purchased from Cell Signaling Technology, and beta-actin (Clone: AC15, Cat#: ab6276, Lot#: GR66278-11) and HRP-conjugated VeriBlot for IP secondary antibody (Cat#: ab131366, Lot#: GR3295852-1) were purchased from Abcam. IRDye 680 conjugated anti-mouse (Cat#: P/N 926-68070) and IRDye 800 conjugated anti-rabbit (Cat#: P/N 926-32211) were obtained from LI-COR Biosciences. Deptor antibody (Cat#: NBP1-49674SS, Lot#: D2) was obtained from Novus Biologicals. Purified anti human HLA-ABC (Clone: EMR8-5, Cat#: 565292, Lot#: 9080800) and HLA-DR, DP, DQ (Clone: Mouse IgG2a,κ, Cat#: 555557, Lot#: 9100932) were purchased from BD Biosciences.

**Cell lines**. Human embryonic kidney HEK293 (Cat#: CRL-1573, ATCC) and HEK293FT (Cat#: R70007, Thermo Fisher Scientific) were maintained in high-glucose Dulbecco Modified Eagle medium (Lonza) containing 10% FBS (Gibco), 1% penicillin-streptomycin (Invitrogen), and L-glutamine (Lonza) in a 37 °C humidified incubator with 5% CO2. The authentication for HEK293 and HEK293FT were performed with GenePrint10 System (Promega). The result was compared to ATCC STR, JCRB STR, ICLC STR database, and DSMZ online STR database. The identity estimates are calculated according to the allele information found in these databases. Mycoplasma test was performed using MycoAlert Mycoplasma Detection Kit (LONZA, Cat#: LT07-318).

**Sample preparation and DNA extraction**. Mononuclear cells (MNCs) were separated from whole blood using Ficoll-Paque™ PLUS (GE Healthcare). The separated MNCs were then labeled with either CD4+ or CD8+ magnetic beads (Miltenyi Biotec) and sorted by AutoMACs® cell sorter (Miltenyi Biotec) according to the manufacturer's protocol. The purity of sorted fractions was evaluated by flow cytometry and confirmed to be >98% (FACSVerse, BD Biosciences). Alternatively, separated MNCs were sorted using FACSAria II (BD Biosciences). Genomic DNA was isolated from fresh or frozen sorted MNCs or from whole blood samples using the Genomic DNA NucleoSpin Tissue kit (Macherey-Nagel). DNA concentration and purity were measured with Qubit2.0 Fluorometer (Invitrogen) or Nanodrop (Thermo Fisher Scientific).

**Flow cytometry analysis and flow-assisted cell sorting**. For phenotyping of the memory T cell subsets, peripheral blood mononuclear cells (PBMCs) were immunostained with the antibody panel including anti-CD3 PeCy7 (Clone: SK7, Cat#: 557851, Lot#: 8037645, BD Biosciences), anti-CD4 PerCP (Clone: SK3, Cat#: 345770, Lot#: 6281605, BD Biosciences), anti-CD8 PerCP (Clone: SK1, Cat#: 345774, Lot#: 82152, BD Biosciences), anti-CD45RA Alexa700 (Clone: HI100, Cat#: 560673, Lot#: 7180940, BD Biosciences), and anti-CCR7 PE (Clone: 150503, Cat#: FAB197P, Lot#: LEU1618031, R&D System). Stained samples were analyzed with FACSVerse (BD Biosciences) and FlowJo software (Version 10.4.2). For CD4+ T cell TCR Vβ20+ bearing clonotype sorting, PBMCs were immunostained with anti-CD3 APC (Clone: SK7, Cat#: 345767, Lot#: 7236657, BD Biosciences), anti-CD4 PerCP (Clone: SK3, Cat#: 345770, Lot#: 6281605, BD Biosciences), anti-CD8 PE-Cy7 (Clone: SK1, Cat#: 335822, Lot#: 8272690, BD Biosciences) and Vβ20 (IOTest® Beta Mark TCR Vbeta Repertoire Kit, Cat#: IM3497, Lot#: 66, Beckman Coulter). Stained cells were physically isolated by FACS AriaIII (BD Biosciences). Purity of sorted cells was more than 99% and verified with the same system. (Supplementary Fig. 2b). For the analysis of Granzyme B+ T cells, PBMCs were stained with anti-CD3 APC (Clone: SK7, Cat#: 345767, Lot#: 7236657), anti-CD45 APCH7 (Clone: 2D1, Cat#: 641417, Lot#: 6265575, BD), anti-CD4 PerCP (Clone: SK3, Cat#: 345770, Lot#: 6281605, BD), and anti-CD8 FITC (Clone: SK1, Cat#: 345772, Lot#: 7235793, BD). Next, cells were fixed and permeabilized using Fix/Perm (Cat#: 554714, Lot#:7346888, BD) and stained using anti-Granzyme B BV510 (Clone: GB11, Cat#: 563388, Lot#: 9093962, BD), anti-TNF-α V450 (Clone: MAb11, Cat#: 561311, Lot#: 8127814, BD) and anti-IFN-γ V450 (Clone: B27, Cat#: 560371, Lot#: 5275825, BD). To stimulate T cells, PBMCs were cultured in RPMI1640 (10% FBS, 1% penicillin/streptomycin, and 2 mM L-glutamin). Stimulation was performed using anti-CD3 APC (Clone: UCHT1, Cat#: 561810, Lot#: 8316946, BD), anti-CD28 (Clone: L293, Cat#: 340975, Lot#: 8277956, BD), anti-CD49 (Clone: L25, Cat#: 340976, Lot#: 5342638, BD). Protein transport inhibitor, GolgiSTOP™ containing monensin (Cat#: 554724, Lot#: 9011506, BD), was added followed by incubation for 16 h at 37 °C.

**Immunopanel sequencing**. A customized NGS panel including exonic areas of 986 genes related to immunity and cancer was used to screen for somatic mutations[12]. Genes included in the panel are provided in the Supplementary Table 7. DNA library were prepared from NEBNext (New England BioLabs, Cat#: E6040L) and ThruPLEX DNA-seq (Rubicon Genomics, Cat#: R400407) according to the manufacturers' instruction. Minor modifications were applied to NEBNext kit: (1) For ligation 1 μL of 25 μM adapter was used with Illumina Index PE adapter oligo mix. Primers were purchased from Sigma Aldrich (5′-GATCGGAAGAGCACACGT CT-3′ and 5′-ACACTCTTTCCCTACACGACGCTCTTCCGATCT-3′). (2) Library purification was proceeded using Agencourt AMPure XP-beads (Beckman Coulter, Cat#: A63881). Preparation of pre-capture PCR was performed with 12 cycles of amplification (15 ng of the ligated library, 5 parallel reactions, initialization: 2 min, denaturation: 20 s). (3) For forward primer we used Illumina PCR Primer InPE 1.0 (5′-AATGATACGGCGACCACCGAGATCTACACTCTTTCCCTACACGACGC

TCTTCCGATCT-3'). Custom made-indexed long reverse primer was merged from the Illumina PCR Primer InPE 2.0 (5'-GTGACTGGAGTTCA-GACGTGTGCTCTTCCGATCT-3') and Illumina TruSeq Small RNA index primer (5'-CAAGCAGAAGACGGCATACGAGATNNNNNNNGTGACTGGAGTTC-3', NNNNNN being the selected small RNA index). NNNNNN sequence from the merged long reverse primer template (5'-CAAGCAGAAGACGGCATACGAGATNNNNNN GTGACTGGAGTTCAGACGTGTGCTCTTCCGATCT-3') was replaced with selected index. For these long reverse primers, the amplification cycle was modified from 19 cycles to 12 cycles. (4) LM-PCR step in the NimbleGen protocol was excluded due to sufficient amount of library. Library quantification was proceeded using Bioanalyzer High sensitivity kit (Agilent, Cat#: 5067-4626).

Target region capture was performed according to NimbleGen SeqCap EZ Exome Library SR User's Guide. Minor modifications were applied to NimbleGen captures: (1) 10 µL of Illumina PCR Primer InPE 1.0, MPLEX_blockAdapter 2.0 (100 µM, 5'-GTGACTGGAGTTCAGACGTGTGCTCTTCCGATCT-3') and MPLEX_blockTail2.0 (100 µM, 5'-CAAGCAGAAGACGGCATACGAGAT-3') were used in each library for blocking. (2) Four microliter of the captured library was used in 5 parallel 50 µL amplification reactions. One microliter of NimbleGen PE-POST1 and PE-POST2 primers (100 µM) were used in each reaction (12 cycles).

ThruPLEX library preparation was performed according to the manufacturer's instruction. Hundred nanogram of gDNA was used to mean fragment size of 300 bps by Episonic Multi-Functional Bioprocessor 1100 (Epigentek Group Inc., NY, USA). The libraries were processed according to ThruPLEX DNA-seq library preparation kit (Rubicon Genomics, MI, USA) with 50 ng of fragmented DNA. The libraries were quantitated with Bioanalyzer 2100 (Agilent Technologies, CA, USA). The captures were performed according to Rubicon Genomics' protocol for Exome Capture of ThruPLEX Libraries with Roche NimbleGen SeqCap EZ Library (Rubicon Genomics, MI, USA). To perform the capture, Roche Nimblegen custom capture probes were used after hybridization.

Library purification for the amplified libraries (pre-capture preparation with NEBNext or ThruPLEX) were performed using Agencourt AMPure XP beads (Beckman Coulter, Cat#: A63881) followed by sequencing quantification using Illumina HiSeq 2500. Sequencing was done from both sorted CD4+ and CD8+ T cells. Somatic variant-calling pipeline was used with VarScan 2.3.2[8,44]. The parameters were set to minimum coverage tumor reads: 6, somatic p value: 1, normal purity: 1 and minimum variant frequency: 0.05. Somatic mutations were annotated using SnpEff (version 4.0) and Ensembl database (version 68)[45,46]. Alignments for the somatic mutations were confirmed in the integrative Genomics Viewer (IGV, Broad Institute, MA, USA). Variants present in dbSNP130 data sets were excluded to filter out misclassified germline variants.

**Exome sequencing.** CD4+ T cells, CD8+ T cells and NK cells of index patient were used for exome sequencing. Fifty nanogram of DNA was used with Nextera Rapid Exome Kit (Illumina, Cat#: FC-140-1083). Somatic variants calling were performed with the same bioinformatics pipeline as with immunopanel sequencing.

**TCR Vβ analysis.** TCR Vβ families were analyzed from peripheral whole-blood samples by flow cytometry-based antibody staining using IOTest® Beta Mark TCR Vβ Repertoire Kit (Cat#: IM3497, Lot#: 66, Beckman Coulter). Briefly, CD4+ and CD8+ T cells in whole blood samples were stained with the panel of TCR Vβ antibodies recognizing 24 members of TCR β chain, which covers about 70% of the normal human TCR Vβ repertoire. Stained cells were further analyzed using FACSVerse (BD Biosciences). The corresponding IMGT gene nomenclature for the vβ segment in the kit is included in the Supplementary Table 8.

**TCR CDR3 deep sequencing.** Isolated genomic DNAs was used for TCRβ deep sequencing. Sequencing and data analysis were conducted by Adaptive Biotechnologies (WA, USA) with ImmunoSEQ assay (Adaptive Biotechnologies, WA, USA)[47]. Briefly, raw sequence reads were demultiplexed and adapter and primer sequences were removed to exclude primer dimer, germline, and contaminant sequences. To merge closely related sequences, data filtering and clustering were performed using the relative frequency ratio between similar clones and a modified nearest algorithm. V, D, and J gene were annotated in accordance with the IMGT database [http://www.imgt.org/].

**Somatic mTOR mutation validation.** A specific primer set was designed using the Primer-Blast search (National Center for Biotechnology Information: http://blast.ncbi.nlm.nih.gov/) to validate the somatic mTOR mutation (Supplementary Table 9). Polymerase chain reaction (PCR) products were purified with the ExoSAP-IT (Affymetrix) followed by sequencing on DNA sequencer (Applied Biosystems). Sequences were analyzed using 4Peaks version 1.7.1.

**Amplicon sequencing of mTOR, NFκB2, and TLR2.** Amplicon sequencing was performed to validate the mutations found in the immunopanel sequencing with the primers (Supplementary Table 10). Amplicons were amplified with 2-step PCR protocol. First PCR was done in a volume of 20 µL containing 10 ng of sample DNA, 10 µL of 2× Phusion High-Fidelity PCR Master Mix and 0.375 µM of each locus-specific primer. The reaction mix was brought to a final volume with water. The second PCR was done in a volume of 20 µL containing 1 µL of the amplified product from the first PCR, 10 µL of 2× Phusion High-Fidelity PCR Master Mix, 0.375 µM of index primer 1 and 0.375 µM of index primer 2. The reaction mix was brought to a final volume with water.

DNA Engine Tetrad 2 (Bio-Rad Laboratories) or G-Storm GS4 (Somerton) thermal cyclers were used to cycle the 1-step PCR samples according to the program: initial denaturation at 98 °C 30 s, 30 cycles at 98 °C for 10 s, at 67 °C for 30 s, and at 72 °C for 15 s, and the final extension at 72 °C for 10 min. Two-step amplifications were done with the same thermal cyclers. The second PCR (Index PCR) was done according to the program: initial denaturation at 98 °C 30 s, 8 cycles at 98 °C for 10 s, at 65 °C for 30 s, and at 72 °C for 20 s, and the final extension at 72 °C for 5 min. The amplified samples were pooled together and the pool was purified with Agencourt AMPure XP beads (Beckman Coulter, CA, USA) twice using 1× volume of beads compared to the sample pool volume. Purification with 1× bead volume removes most of primer dimers from the sample pool. Agilent 2100 Bioanalyzer (Agilent Genomics, CA, USA) was used to quantify amplification performance and yield of the purified sample pools. Sample pools were sequenced with Illumina HiSeq System using Illumina HiSeq Reagent Kit v4 100 cycles kit/MiSeq System using MiSeq 600 cycles kit (Illumina, San Diego, CA, USA).

Sequencing reads alignment was performed with Bowtie2, and GATK IndelRealigner was used for local realignment near indel. The coverage was over 100,000×, and a variant was called if variant base frequency was 0.5% of all reads covering a given a position. All variants with the base quality frequency ratio (ratio of number of variant calls/numbers of all bases and quality sum of variant calls/quality sum of all bases at the position) ≥ 0.9 were considered as true somatic variants[8,12].

**scRNA-seq and TCRαβ-seq processing.** CD4+ T cells from two time points of the index patient were enriched using CD4 microbeads (Miltenyi Biotec). Single cells were partitioned using a Chromium Controller (10× Genomics) and scRNA-seq and TCRαβ-libraries were prepared using Chromium Single Cell 5' Library & Gel Bead Kit (10× Genomics), as per manufacturer's instructions (CG000086 Rev D). In brief, 17,000 cells from each sample, suspended in 0.04% BSA in PBS were loaded on the Chromium Single Cell A Chip. During the run, single-cell barcoded cDNA is generated in nanodroplet partitions. The droplets are subsequently reversed, and the remaining steps are performed in bulk. Full length cDNA was amplified using 14 cycles of PCR (Veriti, Applied Biosystems). TCR cDNA was further amplified in a hemi-nested PCR reaction using Chromium Single Cell Human T Cell V(D)J Enrichment Kit (10× Genomics). Finally, the total cDNA and the TCR-enriched cDNA was subjected to fragmentation, end repair and A-tailing, adapter ligation, and sample index PCR (14 and 9 cycles, respectively). The gene expression libraries were sequenced using an Illumina NovaSeq, S1 flowcell with the following read length configuration: Read1 = 26, i7 = 8, i5 = 0, Read2 = 91. The TCR-enriched libraries were sequenced using an Illumina HiSeq2500 in Rapid Run mode with the following read length configuration: Read1 = 150, i7 = 8, i5 = 0, Read2 = 150. The raw data was processed using Cell Ranger 2.1.1. with GRCh38 as the reference genome.

**scRNAseq analysis of GvHD patient.** Secondary analysis was performed in R with Seurat 3.0.2[48]. In quality control (QC), cells with fewer than 200 or more than 4000 genes, more than 15% of the counts from mitochondrially-encoded transcripts or did not have TCRα, TCRβ, or TCRαβ were excluded from the analysis. The remaining data was log-normalized with scaling factor 10,000. To reduce the dimensionality of the data, we determined the 1000 most highly variable genes with the FindVariableFeatures-function with "vst" method. The T cell receptor V(D)J-genes, mitochondrial genes and ribosomal genes (n = 156) were excluded from the results. We scaled the data based on all genes and performed PCA on the highly variable genes and the top principal components (PCs) that had standard deviation higher than 1.5 were kept (n = 10). Clusters were identified using the graph-based community identification algorithm from the shared-nearest neighbor graph calculated on the PCs as implemented in the Seurat-package. To prevent over-clustering, the optimal number of clusters was determined by increasing the resolution hyperparameter as a function of number of clusters until the first saturation plateau was achieved (in our case resolution = 0.4). The robustness of these clusters was assessed by subsampling cells and doing the analysis iteratively and visually inspecting the results of embedding and differentially expressed genes between the formed clusters. Differential expression analysis was performed based on the t-test, as suggested by Robinson et al.[49] Clusters were annotated using canonical cell type markers as well as the differentially expressed genes. Gene Set Enrichment Analysis (GSEA) (software.broadinstitute.org/gsea/index.jsp) between the clonotype and other cytotoxic cells was performed on genes that were detected at least in 0.1% of the cytotoxic cells and had at least log fold-change of 0.01 between the clonotype and other cytotoxic CD4+ T cells. The gene list was ordered based on the fold-change. Overlap with HALLMARK-category was assessed and the False Discovery Rate (FDR) calculated while the number of permutations was 1000. Clonotypes were identified based on the available information and both total nucleotide level TCRα and TCRβ were used if found.

**scRNAseq analysis of healthy donor**. To compare our findings of CD4[+] T cells from GvHD patient to healthy donor, we downloaded single-cell and protein-level data of 5247 peripheral blood mononuclear cells from a healthy donor from the 10× website (https://support.10xgenomics.com/single-cell-gene-expression/datasets/3.1.0/5k_pbmc_protein_v3). The analysis followed the vignette found on the Seurat webpage (https://satijalab.org/seurat/v3.1/multimodal_vignette.html). Cells with less than 200 genes were removed from the analysis. We performed clustering directly on protein levels with shared-nearest neighbor graph and clustering resolution parameter 0.2 We identified 11 clusters and proceeded to focus on the two CD4[+] T cell clusters, naïve and memory CD4[+] T cells. For these, we performed dimensionality reduction on RNA-level to visualize the expression of cytotoxic markers.

**TCRαβ analysis**. From the TCRβ-seq we detected TCRαβ, TCRα or TCRβ from 14,111 cells, resulting in 3578 different T cell clonotypes. The Vβ20 bearing clonotype that harbor the *mTOR*-mutation was the most expanded, compromising of 4669 cells (33.10%, TRA: CLVGDIGNQGGKLIF; TRB: CAWSTGQANNSPLHF). However, we noticed that the second most (TRB: CAWSTGQANNSPLHF, 2038 cells, 14.40%) clonotype had only one chain and as its Vβ, Dβ, and Jβ matched to the most expanded clonotype, we treated this as error coming from uncomplete sequencing and pooled the two most expanded clonotypes into one.

**Analysis of cellular cytotoxicity**. Primary fibroblasts from the index patient and three healthy donors were cultured. Briefly, skin biopsy from the index patient were dissected in small pieces (approx. 2 mm × 2 mm) and transferred into 6-well plate in 500 μL of complete growth medium containing 20% FBS. 200–300 μL of growth medium was added for every 2 days to replace evaporated media. After one week, we increased the amount of media to 2 mL and changed the media every 3 days. Once cells were confluent in each well, cells were trypsinized and passaged.

To measure cellular cytotoxicity of CD4[+] and CD8[+] T cells, the proliferation of the fibroblast was monitored with xCELLigence[TM] real-time cell analyzer (RTCA) (ACEA Biosciences, CA, USA) according to the manufacture's instruction. xCELLigence[TM] RTCA biosensor measures cellular adhesion through electrical impedance, which is converted to Cell Index (arbitrary units). CD4[+] T cells and CD8[+] T cells were separated from MNCs with CD4[+] or CD8[+] magnetic beads (Miltenyi Biotec) and sorted by AutoMACs® cell sorter (Miltenyi Biotec) according to the manufacturer's protocol. Briefly, the E-Plate 16 VIEW (ACEA Biosciences, CA, USA) was equilibrated with the 100 μL of culture media at room temperature. Next, 100 μL of the cell suspension (fibroblasts, $8 \times 10^3$ cells/well) was transferred to the plate followed by incubation at room temperature for 30 min to allow the fibroblasts to settle at the bottom of the wells. The xCELLigence[TM] monitored the cells every 30 min for 200 repetitions. When the cell index was reached a plateau, CD4[+] T cells, CD8[+] T cells, and NK92 as a cellular cytotoxicity inducer ($6.4 \times 10^4$ cells as a ratio of the fibroblast to the inducer is 1:8) were added to the fibroblasts. The real-time impedance trace for the fibroblasts exposed to CD4[+] T cells, CD8[+] T cells, and NK92 were monitored for 48 h.

The cell index (CI) was calculated according to the formula from RTCA software (xCELLigence[TM]):[50]

$$CI(t) = \frac{R(f_n, t) - R(f_n, t_0)}{Z_n},$$

where $f_n$ is the frequency that impedance measurement is carried out. $R(f_n, t)$ is the measured impedance at frequency $f_n$ at time point $t$. $R(f_n, t_0)$ is the measured impedance at frequency $f_n$ at time point $t_0$ (usually $t0$ is the time when the background is measured). $Z_n$ is the corresponding frequency factor of $f_n$.

Normalized cell index (NCI) which is normalized from the cell index of different wells was calculated with the following formula:

$$NCI_{well\_i}(t) = \frac{CI_{well\_i}(t)}{CI_{well}(t_{normalization})} \text{ and } CI_{well\_i}(t_{normalization}) \neq 0,$$

$NCI_{well\_i}(t)$ is the normalized cell index of well i at time point $t$. $CI_{well\_i}(t)$ is the Cell Index of well i at time point $t$. $CI_{well\_i}(t_{normalization})$ is the Cell Index of well i at the normalization time $t_{normalization}$.

**Multiplexed immunohistochemistry (mIHC)**. Tissue blocks were cut in 3.5 μm sections. Slides were deparaffinized in xylene and rehydrated in graded ethanol series and $H_2O$. Heat-induced epitope retrieval (HIER) was carried out in 10 mM Tris-HCl–1 mM EDTA buffer in +99 °C for 20 min (PT Module, Thermo Fisher Scientific). Peroxide activity was blocked in 0.9% $H_2O_2$ solution for 15 min, and protein block performed with 10% normal goat serum (TBS-NGS) for 15 min. Anti-CD3 (Clone: EP449E, Cat#: ab52959, Lot#: GR140731, Abcam) primary antibody diluted 1:500 in protein blocking solution and secondary anti-rabbit horseradish peroxidase-conjugated (HRP) antibodies (Immunologic) diluted 1:1 in washing buffer were applied for 1 h 45 min and 45 min, respectively. Tyramide signal amplification (TSA) 488 (PerkinElmer) was applied on the slides for 10 min. Thereafter, HIER, peroxide and protein block were repeated, followed by application of anti-CD8 (1:500, Clone: C8/144B, Cat#: BSB 5174, Lot#: 5174JDL05, BioSB) primary antibody, HRP-conjugated secondary antibody diluted 1:3 with washing buffer and TSA 555 (PerkinElmer). HIER, peroxide block and protein

block were repeated. Then, the slides were incubated with CD4 primary antibodies (1:25, Clone: EPR6885, Cat#: ab133616, Lot#: GR218457, Abcam) overnight in +4 °C. Next, AlexaFluor647 fluorochrome-conjugated secondary antibody (Thermo Fisher Scientific) diluted in 1:150 and Dapi (Roche) counterstain diluted 1:250 in washing buffer were applied for 45 min. ProLong Gold mountant (Thermo Fisher Scientific) and a coverslip were applied on the slides. After peroxide block, antibody incubations and fluorochrome reaction, slides were washed three times with 0.1% Tween-20 (Thermo Fisher Scientific) diluted in 10 mM Tris-HCL buffered saline pH 7.4 (TBS). Fluorescent images were acquired with the AxioImager.Z2 (Zeiss) microscope equipped with a Zeiss Plan-Apochromat 20× objective.

**Site-directed mutagenesis**. Site-directed mutagenesis was conducted using GENEART® Site-Directed mutagenesis system according to the manufacturer's instruction (Invitrogen) with *NFκB2* (GeneCopoeia, Cat. EX-Z4293-Lv154), *TLR2* (GeneCopoeia, Cat. EX-Q0161-Lv122), and *mTOR* (Addgene, Cat. 26603) expression vector. The primer sequences used for the site-directed mutagenesis are in the Supplemental Table 11.

**Establishing stable cell lines**. HEK293 cells were transfected using FuGENE HD transfection reagent (Promega) with either a wildtype or P2229R *mTOR* expression vector (ratio of reagent to DNA is 3:1) following the manufacturer's instruction. Neomycin resistant clones were selected after the cells were cultured with G418 (500 μg/mL) for 3 weeks. The lentiviruses were produced by co-transfection of HEK293FT cells with *NFκB2* (wildtype or P882Q mutant) or *TLR2* (wildtype or W558L mutant) lentiviral expression vectors, and psPAX2 lentiviral packaging plasmid (Addgene) and pCMV-VSV-G envelope plasmid (Addgene) using Lipofectamine® 2000 (Thermo Fisher Scientific). Antibiotic-free DMEM containing 10% FBS was used as a culturing medium and Opti-MEM I Reduced Serum Medium (Thermo Fisher Scientific) supplemented with 5% FBS and 1 mM Sodium pyruvate was used as a lentivirus packaging medium. Six hours of post-transfection, medium was removed and replaced with DMEM. After 48 h, the supernatants were centrifuged at 300×g for 5 min to remove cell debris and filtered with a 0.45 μm polyethersulfone membrane filter. Ultracentrifugation to concentrate the virus was performed for 2 h at 12,000×g and 4 °C using Beckman SW28 rotor. Lentivirus titers were measured by p24 specific enzyme-linked immunosorbent assay.

**Establishment of triple mutant stable cell lines**. HEK293 cells stably expressing exogenous *mTOR* (wildtype or P2229R mutant) were transfected with *NFκB2* (wild type or P882Q mutant) expressing lentiviruses. Infections were performed in the presence of 8 μg/mL of polybrene under centrifugation (500×g, 37 °C) for 2 h. *mTOR-NFκB2* transduced cells (expressing Cyan Fluorescent Protein) were selected by using FACSAriaIII (BD Biosciences). Cells expressing exogenous *mTOR-NFκB2* (wildtype or mutant) were infected with *TLR2* (wildtype or W558L mutant) expressing lentiviruses as described above and selected using puromycin (3 μg/mL).

**Western blot analysis and immunoprecipitations**. HEK293 cells were washed twice with cold PBS followed by serum starvation for 12 h. Cells were then harvested and further lysed in ice-cold RIPA buffer with 1× protease and phosphatase inhibitor cocktail (Thermo Fisher Scientific). To remove cell debris, centrifugation was carried out for 10 min at 4 °C, 12,000×g. Total protein concentration was measured with the Qubit protein assay (Thermo Fisher Scientific) and protein samples were prepared in Laemmli buffer (Bio-Rad Laboratories) to load on SDS-PAGE gel (Bio-Rad). After running the sample in the SDS-PAGE gel, the proteins were transferred into a nitrocellulose membrane using Trans-blot Turbo transfer system (Bio-rad) followed by blocking the membrane with Odyssey blocking buffer (Cat: 927-40000, LI-COR Biosciences) for 1 h. Primary antibodies (1:1000 dilution) were incubated overnight at 4 °C in the Odyssey blocking buffer with 0.2% Tween 20, and subsequently secondary antibodies (1:15,000 dilution) in the blocking buffer with 0.2% Tween 20 were incubated for 1 h at room temperature. The proteins were visualized using Odyssey Imaging Systems (LI-COR Biosciences). For co-immunoprecipitation, cells were lysed using Pierce[TM] IP lysis buffer (Cat#: 87788, ThermoFisher). One milligram of total cell lysates were pre-cleared with PureProteome™ Protein A/G Mix Magnetic Beads (Cat#: LSKMAGAG02, Merck) for 1 h at 4 °C. Pre-cleared supernatants were incubated with anti-mTOR antibody for overnight at 4 °C under rotation. Magnetic beads were additionally added to the immune complexes, and then incubated at room temperature for 15 min under rotation. The immune complexes were washed three times in PBST, resuspended in the elution buffer and heated at 70 °C for 10 min. Bands were quantified using ImageJ and uncropped scan of the blots are in the Source data file.

**Quantitative reverse transcription PCR (RT-qPCR)**. Total RNA was extracted using the RNeasy Mini kit (Qiagen) followed by cDNA synthesis using QuantiNova Reverse Transcription kit (Qiagen) according to the manufacturer's protocol. The cDNA was applied in SYBR Green RT-PCR master mix (Applied Biosystems) and oligonucleotide primers (Supplementary Table 12). All RT-qPCR reactions were performed in 384-microwell plates (Applied Biosystem) using a QuantStudio 6 Flex Real-Time PCR system (Applied Biosystems). The relative quantitation of gene expression was analyzed using comparative cycle threshold (ΔΔCT) method, and

beta actin (ACTB) was used as an endogenous control to normalize gene expression level.

**Cell proliferation assay**. Cells were seeded in a 96-well plate (1000 cells/well) in triplicate. Cell proliferation was determined using CyQUANT™ Cell Proliferation Assay which is similar to the high-content screening assay in sensitivity following the manufacturer's instruction (Cat#: C35011, Invitrogen). Briefly, at indicated time point, 100 μL of detection reagent combined with nucleic acid stain dye and background suppressor was added to 100 μL of cells in the plate followed by incubation for 1 h at 37 °C. Fluorescence was measured using FITC filter. Cell numbers were calculated based on the standard curve according to the manufacturer's protocol.

**Drug sensitivity and resistance testing (DSRT)**. Ex vivo DSRT was performed on freshly isolated CD4+ T cells with a total of 527 drugs in five concentrations covering a 10,000-fold concentration range including conventional chemotherapeutics and a broad range of targeted oncology compounds[51]. To dissolve the drug compounds, 5 μL of medium was dispensed into each well of 384-well plates including five different concentrations of each drug. Twenty microliter of cell suspension (CD4+ T cells from healthy control, HSCT donor, and index patient: 2000 cells per well) was transferred to every well using MultiFlo FX dispenser (BioTek). After incubation (5% $CO_2$ at 37 °C) for 72 h, the cell viability was evaluated by CellTiter-Glo Assay solution (Promega). The drug sensitivity score (DSS) was calculated to evaluate quantitative drug profiles based on the measured dose-response curve[52].

**Statistical analysis**. Unpaired two side $t$-tests and Mann–Whitney test was performed using GraphPad Prism 8 for Mac OS X (version 8.2.1). Multiple test correction was assessed with Benjamini-Hochberg method implemented in based R for Mac OS X (version 3.5.1). $p$-value < 0.05 was considered as statistically significant in all analyses as indicated in the figure legends and methods. Normalization of the cell index from the real-time monitoring of cellular cytotoxicity was performed using xCELLigence RTCA Software (version 2.1.0, ACEA Biosciences, Inc.).

**Reporting summary**. Further information on research design is available in the Nature Research Reporting Summary linked to this article.

## Data availability

Data supporting the findings of this work are available within the paper and its Supplementary Information files. The source data underlying Figs. 1b, d, 2c, 3a, d, e, 4, 5a, c, d, 6a–g, and Supplementary Figs. 3, 4, 5b, 6b, 9a–d, 10a, b, and 11 are provided as a Source Data file. The TCRB sequence data that support the findings of this study are available through the ImmuneAccess platform [https://doi.org/10.21417/DK2020NC] and also in the Supplementary Data 1 ZIP file. The scRNA-seq from healthy control is available from [https://support.10xgenomics.com/single-cell-gene-expression/datasets/3.1.0/5k_pbmc_protein_v3]. The read counts of scRNA-seq data from index patient have been deposited in the ArrayExpress database at EMBL-EBI (www.ebi.ac.uk/arrayexpress) under the accession number E-MTAB-8911. Somatic variants both from whole exome sequencing (index patient) and and amplicon sequencing (GvHD patients and healthy controls) have been deposited in dbSNP (ss2137544086, ss3983910085, ss3983910086, ss3983910087, ss3983910088, ss3983910089, ss3983910090, ss3983910091, ss3983910092, ss3983910093, ss3983910094, ss3983910095, ss3983910096, ss3983910097, ss3983910098, ss3983910099, ss3983910100, ss3983910101, ss3983910102, ss3983910103, ss3983910104, ss3983910105, ss3983910106, ss3983910107, ss3983910108, ss3983910109, ss3983910110 [http://www.ncbi.nlm.nih.gov/SNP/snp_viewTable.cgi?handle=HRUH_MUSTJOKI].

## Code availability

All custom scripts made for scRNA-seq, TCRαβ-seq and healthy data are available at [https://github.com/janihuuh/gvhd_som_mut].

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

## Acknowledgements

This work was supported by the European Research Council (M-IMM project), Academy of Finland (Decisions 287224, 314442), Finnish special governmental subsidy for health sciences, research and training, Sigrid Juselius Foundation, Instrumentarium Science foundation, Helsinki Institute of Life Sciences Fellow funding, Cancer Foundation Finland, and Finnish Cancer Institute. T.L. was supported by the Academy of Finland (Decisions 311081 and 314557). This study was supported by Finnish Functional Genomics Center, University of Turku, Åbo Akademi University and Biocenter Finland. IT Center for Science LTD (CSC) is acknowledged for their help and computing resources.

## Author contributions

D.K., G.P., S.L., R.K., M.M., M.A.I.K., and S.M. designed the study and experiments. A.M.H., C.M-C., L.C., V.G.G.S., T.H.C-L., A.K., A.J., U.S., M.A.I.K., and M.I-R. contributed and prepared biological samples and clinical data. D.K., G.P., S.L., R.K., O.B., J.H., M.M., and T.L. performed experiments and analyzed the data. P.E. and S.H. designed, supervised and performed sequencing assays. S.E., J.H., and M.K. performed bioinformatic analyses. S.M. conceived and designed the study, directed and supervised research. D.K, G.P, M.M, M.A.I.K., and S.M. wrote the manuscript. All authors read and approved the final manuscript.

## Competing interests

S.M. has received honoraria and research funding from Novartis, Pfizer and Bristol-Myers Squibb (not related to this study). The remaining authors declare no competing interests.
