## [Peer Review File · Nature Communications]

Reviewers' comments:

Reviewer #1, clinical GvHD expert (Remarks to the Author):

This is an extensive study that characterizes a gain of function somatic mutation in MTOR in 3 cGVHD patients. They identify an expanded CD4+ T cell clone in the index patient at two different time-points, in blood and tissue (skin), demonstrate the absence in T cells directly attained from the donor, show functional validity against the patient's fibroblasts and suggest they may be resistant to mTOR inhibitors but susceptible to HSP90 inhibition. The experiments are well-done, but the implications/conclusions need to be tempered.

The issues are:

1. The generalizability of this specific somatic mutation occurrence is very low (only 3/135 cGVHD) thus the significance for cGVHD patients as a whole is small. Second, the patients and the control cohorts are extremely heterogeneous, for the type, age, cGVHD status, treatment status, time points of analyses, lack of donor analyses, the places of transplant- all of which make the conclusion that these mutations may be valid markers for cGVHD not particularly robust. Would suggest toning down of any discussion/ implication as a potential for cGVHD patients in the title and through out the manuscript.
2. The value of the other two mutations (NFkB and TLR2), even though mentioned in abstract, introduction and esp NFkB in discussion, is neither clear nor validated. Would recommend tempering the implications of those mutations throughout the manuscript.
3. In a similar note- the discussion throughout is extremely speculative including (not limited to the DUSP2, NFkB) and their implications. Would need to appropriately reduce and provide a more balanced context and content.
4. cGVHD is complex and manifestations and treatments myriad from patient to patient and across physicians and centers. Please provide in greater detail the index and the other cGVHD patients clinical profile and treatment context in greater detail. The index patient- was the person on sirolimus anytime before or during or after the time points of analysis. If cGVHD intensity increase, wax or wane or was it stable for the 2 years? If so how would the authors interpret the putative causal role of the CD4 clone to the changes in the disease? More details and also discussion on the level of immunosuppression (types and intensity of various drugs including steroids, CNIs etc) should be provided.
5. For the CTL assays with CD4+ T cells- is provocative. What were the negative target controls for the CD4 T cell clones?
Minor:
6. Some of the references do not seem to match the place where cited- too many to be specific, so pl cross-check them all.

Reviewer #2, expert in T cell repertoire (Remarks to the Author):

The paper by Park and coworkers describes somatic mutations in a chronic GvHD patient potentially related to the GvHD. The analysis is impressive and the hypothesis regarding acquired somatic mutations driving cGVHD is provocative.

Major comments

A major drawback is that the findings presented are confined to a single patient, making it however a very thorough case report. Only the identified mutations, but not their effect, are evaluated in a larger number of patients. Nonetheless the authors extrapolate to all cGVHD patients. The report is a case report at heart, and the broader value is uncertain given the prevalence of the mutations in cGVHD patients.

The findings are interesting, but more work (samples) is needed to elevate the report above anecdotal findings.

In terms of occurrence the TLR2 mutation seems much more interesting than the mTOR - the focus of the manuscript. However, the true merit of the paper lies in identification of the CD4+ T-cells in

cGvHD as being highly cytotoxic. This should be evaluated for the remaining 134 patients, the non-cGvHD patients and the healthy controls. The real-time cytotox assay is attractive for its functionality, but should be followed up by gene analysis where real-time PCR on bulk CD4 and CD8 will suffice. Given higher prevalence of the TLR2 mutation in cGvHD than in healthy controls, an interesting question is if this mutation is associated with, or even predictive for, cGvHD.

The abstract makes the reader believe the body of work is carried out on a cohort of 135 patients - this is not the case. The title is also misleading and the constant extrapolation from one patient to all cGvHD patients is exhausting.

The presented evidence does not support the notion that the mutations were acquired after the transplantation: The authors compare mutations in a T cell population with clearly expanded clones to a population of unknown clone distribution, making it possible that the original clone with the mutation is in such a low frequency it avoids detection in the donor. The apparent increase in the cells harboring the mutations in at least mTOR seems to be increasing with time.

Do the T-cells using the Vb20 also increase over time as does the mTOR mutation? Do other cGvHD patients have expanded clonotypes?

The mTOR P2229R mutation is found in 2.2% of the evaluated cGvHD patients. What about the remaining 97.8% - do they have other mutations in mTOR or TLR2? Do the three patients have other things in common other than the mTOR P2229R mutation?

The pooling of TCRs - where the beta-chain or alpha-chain were missing, but the present counterpart had identical CDR3 to an expanded TCRab set - assuming identical source is problematic. Is there further evidence that the events should be pooled - other than the CDR3?

The TCR nomenclature should be unified. The 'IOTest Beta Mark TCR V β Repertoire Kit' makes use of a variation of the Arden nomenclature, while Adaptive Biotechnologies have their own variation of the IMGT nomenclature. I suggest sticking with the IMGT nomenclature, but any one is possible as long as it is coherent and clearly stated in the materials and methods.

The presentation of the discovery of the mTOR, NF κ B2, and TKR2 mutation and their accumulation is confusing. Their association to Vb20 is also not made clear. As this is a supporting pillar of the manuscript it should be rectified.

Figure 3: It is not necessary to evaluate the effect of the NF κ B2 mutation given that this mutation is confined to the index patient. It would be great to see the effects of the mTOR mutation alone, the TLR2 mutation alone, and the two combined. At least the individual effects of the mutations should be demonstrated and not just the combined effect.

Figure 4:

It is unclear what is being presented exactly. CD4+ cells from 2015 and 2017 were sequenced, but only one set of data is shown. The supplement figure S5 show a less than 2 fold change in overall gene expression between the two time points as argument for pooling, but afterwards the mTOR mutation. Given an increase in VAF of the mTOR mutation over time, the two samples should be split - at least to demonstrate that they do not differ dramatically and can be pooled for further analysis.

The criteria for 'similar phenotype' used in D and E are elusive and should be clearly defined.

It is extremely surprising that the majority CD4+ T-cells form a cluster which can be termed cytotoxic. When zooming into figure 4B it becomes evident that among others granzymes and perforin help to define the clusters as cytotoxic. Only traces of CD8 was found in all clusters indicating that a vast majority of the CD4 T-cells in this cGvHD patient are cytotoxic. The authors, however, does not seem to be baffled by this extremely counter-intuitive observation. More thoughts and observations devoted to this observation is needed.

A log₂(FC) of 0.25 is not dramatic (1.2 times more (FC) in one compared to the other population) and the expression of only two genes is half or double that of the reference population. To verify this is indeed associated with the Vb20 mTOR-mut sub-population and above the noise level, the observations should be compared to repeat random sampling of all the cells in the cytotoxic cluster.

Minor comments

I suggest the use of the word 'AIRR-seq' (adaptive immune receptor repertoire) or 'TCR AIRR-seq' in place of TCRab-seq.

It would be interesting to see plots for NF κ B2 and TLR2 similar to mTOR in figure 2C, where the increase of mutation burden is evident. Given that also other mutations were identified in CD4+, their

development over time would also be interesting to see.

There are quite a few imprecisions and grammatical errors - a few are mentioned below though the list is not exhaustive.

The two chains of the TCR are termed α and β , please stick to this notation and avoid Vb, TCRa, TCRb, and TCRab

The identified expanded Vb20 bearing clonotype is usually referred to as 'the clone' or 'the clonotype'. A more precise notation in order and could be 'Vb20 bearing clonotype' though other possibilities also exist

The discussion skips on the ongoing debate of cytotoxic CD4+ T cells in GvHD. One paper that comes to mind is Du et al, J Immunol., 2015 (doi: 10.4049/jimmunol.1500668) but there are others equally relevant.

Given that some plots are non-Graphpad Prism, the appropriate sources should be cited

Abstract

Lines 51-52: This is a clear overstatement and should be toned down dramatically

Introduction

Line 59: Give frequency rather than just 'in many patients'

Line 61: Please insert reference (not for the organs, but for the lymphocytes bearing the driver)

Line 69-70: An overstatement; the mechanism - host derived antigen - is clear and eloquently described in the beginning of the paragraph.

Results

Line 91: Given that only the V-gene usage is assessed, it is misleading to speak of clones. More importantly, 'clonality' is a very loosely defined concept within T-cell biology owing to the complete lack of a universal formal definition and should always be avoided; or at the least formally defined.

Line 94: To the best of my ability, I cannot see Vb20 in CD8+ in Fig 1B. The purpose of Fig 1A is elusive since TCR V-beta for CD8 is not shown. The purpose of showing Total CD4 and CD4+CD8+ without further mention is equally elusive.

Line 96: See comment for Line 91

Line 100: How do these two V-genes translate to the flow cytometry results?

Lines 101-108: The value of the information is unclear. There is no information to extract from a comparison to the HSCT-donor, rather a before-after the flare comparison would be more appropriate. The value of such information in the context of the paper remain unclear

Line 144: I fail to find the VAF information behind figure 2C - I am only able to find CD4+ from 2015 in Table 2. The tabular summaries are great. Given that there was refinement of the question over time, it would be nice to see the summaries for all compared samples (for instance CD4+ from 2013, 2015, and 2017 in the same table)

Line 146: Incorrect figure reference

Lines 153-158: Are Vb20 CD4 T-cells only present in skin lesions? Maybe the mTOR mutation is inconsequential and their expansion is due to antigen in the skin.

Line 166: Indication that the index is part of the 135 patient cohort. If the total number of all patients with cGvHD 135 or is it 136? If it is 135, it is more prudent to use indicate the validation cohort to be of 134 cGvHD patients

Materials and methods

The index patient is not mentioned. Was the patient in a separate cohort?

Line 375: Please add 'allo-HSCT'

Line 392-393: Missing clone and cat#

Line 399: Duplicate and misleading information; by reading it becomes clear that HEK293 and not HEK293FT were verified

Line 400: Missing preposition

Line 427 and elsewhere: FACS not FACs

Lines 480-496: It is not clear that the analysis is performed in R, it is not clear how the dimensionality is reduced. Generally it is hard to follow the analysis.

Line 502-504: The first sentence is nonsensical. It is not clear what 'two recombinations' entail and hints that commonly occurring T-cells using the same beta-chain with two different alpha-chains are excluded.

Line 516: Should probably read 200-300µl

Line 517: Switch from past to present tense

Line 528: Maybe write as a proper equation

Lines 524-534: It is unclear how long the fibroblasts were allowed to settle

Line 605: It is not clear if cells from two or three donors were used in the screen

Discussion

Lines 262-263: Clear overstatement. It is really hard to extrapolate from 2.2% to all cGvHD patients.

Line 263: That the mTOR mutation is limited to the expanded Vb20 T-cell clone is not formally shown.
Figures and tables

Table 1: Header: Is a minor annoyance to see the word 'tumour' in relation to the analysis of bulk CD4+

Line 859: 'expanded CD4+' is misleading. According to the material and methods as well as the main text it is all CD4+ T cells that were evaluated.

Figure 1A: The plot annotation is too small to be read. Unclear what TCR V-beta FITC and TCR V-beta PE represent and this is not explained in the figure caption.

Figure 1B: The axis labels and titles are too small to be read. The plot would benefit from a clear indication of the information (CD4 or CD8) as a title, in the axis title, or both

Figure 1C: The axis labels and titles are too small to be read. 3D plots should be shunned, and could be represented in a heatmap or - given the few number z-axis values - in individual plots

Figure 1D: The plot annotation is too small to be read.

Figure 2D: The nucleotide calls are too small to be read.

Figure 2C: The colouring is too weak to be seen clearly and the indicator arrows are white blobs - especially in a print out. What are the two images? The caption gives no hint. From the main text it becomes clear that the aim is to underline T-cell infiltration in the skin lesion. Can this be quantified? Can this be compared to infiltration in the affected eye and liver, and directly compared with the VAF? As it is now, the information is in three places: text, table and figure. If possible it should be combined
Figure 3B-3D: The axis labels and titles are too small to be read. The bars indicate SEM. Unless each point is the mean of a number of samples (in which case the numbers should be added), SEM is the wrong statistics and should be replaced by SD

Figure 4: All axis labels and titles are too small to be read.

Figure 4D: The plot make use of two different red, but the caption mention red and gray. There should be a legend on the plot

Figure 4E: The non-red colour is practically invisible when printed. What does that colour represent? There should be a legend on the plot. Does the y-axis really represent log 100 of the p-value?

Figure 4E: The resolution is too low and the annotation too small

Figure 6: All axis labels and titles are too small to be read.

Figure 6B; 6D: axis title and labels are missing for the x-axis

Figure 6F; 6G: See comment for Figure 3B-3D

Supplementary

Page numbers missing in table of contents

Figure S2: What is 'donor' exactly? Only by reading the main text does it become clear that it is the HSCT-donor and not some healthy blood donor

Figure S3: Different symbols are fine, but please use x-axis labels.

Figure S4A: The evaluation of expression levels should be shown for all introduced constructs.

Figure S4B: This is relevant enough to be in the main figure

Primer sequences of Table S3, S5, S6 cannot be extracted directly, they should be readily available.

Reviewer #3, expert in mTOR pathway (Remarks to the Author):

The authors identified persisting somatic MTOR, NFKB2, and TLR2 mutations in an expanded CD4+ T clone. The MTOR P2229R kinase domain mutation was detected in two additional cGvHD patients, but not in controls. Functional analysis of the discovered MTOR mutation indicated a gain-of-function

alteration in translational regulation yielding in up-regulation of phosphorylated S6K1, S6, and AKT. Paired single-cell RNA and TCR sequencing revealed cytotoxicity and abnormal proliferation of the clonally expanded CD4+ T cells. The results are interesting but need additional support. Most importantly, the authors need to prove that MTOR P2229R mutation is a truly gain-of-function by introducing such mutation into wild-type cells, and test the functional outcome (also see below). Without such data, the observed phenotypes could be entirely secondary to alterations in other genes or pathways. The authors need to perform additional assays related to MTOR signaling, including p-4EBPs, p-Foxo1, cell proliferation, cell death, cytokine production, etc. Finally, the authors need to carefully analyze the effects of MTOR P2229R on mTORC1 and mTORC2 pathways.

Response to the reviewer comments

Manuscript Number: NCOMMS-19-27299.

Title: Somatic Mutations in Clonally Expanded T-lymphocytes in Patients with Chronic Graft-Versus-Host Disease

Revised title: Somatic Gain-of-Function *mTOR* mutation in Clonally Expanded T-lymphocytes Associated with Chronic Graft-Versus-Host Disease

We have modified and improved the manuscript according to the reviewers' comments. Main modifications are summarized below followed by detailed point-by-point responses.

- We have edited the text to temper the conclusion and discussion about the generalizability of somatic *mTOR* mutations to all cGVHD patients. In addition, we have modified the title (requested by reviewers 1 and 2).
- We have performed detailed functional validation of the *mTOR* P2229R mutation using HEK293 cell lines stably expressing wild type or mutant *mTOR* (requested by reviewer 3). The results are shown in Figure 3 and in supplementary figures S4-S5. Overexpression of the mutant protein conferred increased cell proliferation and decreased apoptosis upon serum starvation, which was associated with activation of both mTORC1 and mTORC2 pathways. These data strongly suggest gain-of-function effect of this mutation.
- We have reduced the implications of *NFKB2* and *TLR2* mutations in both results and discussion, and we have moved functional studies of these mutations to the supplementary Figure S6 (requested by reviewer 1 and reviewer 2).
- We have included the clinical characteristics of the index patient and two additional cGVHD patients carrying *mTOR* mutation in supplementary results (requested by reviewer 1).
- We have performed additional cytotoxicity assays using T cells from healthy controls as a negative control. Additionally, we have included data showing that blocking MHC class I with an antibody attenuates the cytotoxicity of CD4⁺ T cells, implicating abnormal recognition of MHC class I by CD4⁺ T cells during cGVHD. Furthermore, we now demonstrate dose-dependent cytotoxicity of CD4⁺ T cells in the cytotoxicity assay. These data are presented in Figure 5 and supplementary Figure S9 (requested by reviewer 1).
- We have performed granzyme B flow cytometry analysis to investigate the granzyme B population and Th-1 cytokine production in a cohort of cGVHD patients and healthy controls that we were able to use in the study. These data are added in Figure 1f and supplementary Figure S8 (requested by reviewer 2).
- We have included index patient's amplicon sequencing data and Vβ20 expansions from 2013, 2015, 2017 and 2019. This data is presented in Figure 2c and supplementary table S3 (requested by reviewer 2).
- We have modified the pooling of mutated clonotype to include two of biggest clonotypes with same Vβ, CDR3b and Jb genes in both single cell analysis and TCRβ data from Vβ sorted clonotype (requested by reviewer 2).
- We have included single cell analysis data from a healthy control to demonstrate the cytotoxic markers is limited in the index patient's CD4⁺ T cells in Figure 4c-d. Github page has been created including code used in sc RNAseq analysis:
- https://github.com/janihuuh/gvhd_som_mut
- Due to the revision, we have added new supplementary Figures (2c, 4, 5, 7, 8, 9, 10) in addition to the previous ones (Fig. S6_{new}= Fig. 3_{old} and Fig. S4a-b_{old}, Fig. S7_{new}= Fig. S5_{old}, Fig. S11_{new}= Fig. 6f, g, i_{old}). We have included new supplementary tables (5, 8) in addition to the previous ones (1_{new}= 7_{old}, 2_{new}= 8_{old}, 3_{new}= 9_{old}, 4_{new}= 10_{old}, 6_{new}= 1_{old}, 7_{new}= 2_{old}, 9_{new}= 3_{old}, 10_{new}= 4_{old}, 11_{new}= 5_{old}, 12_{new}= 6_{old}).

Revised portions in the manuscript text, supplementary figures, supplementary results and supplementary tables have been highlighted with yellow colour for visibility.

Reviewer #1, clinical GvHD expert (Remarks to the Author):

This is an extensive study that characterizes a gain of function somatic mutation in MTOR in 3 cGVHD patients. They identify it an expanded CD4+ T cell clone in the index patient at two different time-points, in blood and tissue (skin), demonstrate the absence in T cells directly attained from the donor, show functional validity against the patient's fibroblasts and suggest they may be resistant to mTOR inhibitors but susceptible to HSP90 inhibition. The experiments are well-done, but the implications/conclusions need to be tempered.

#We thank the reviewer for the positive feedback on our study. We have revised the text and tempered the conclusions as suggested. In the discussion, we have added a separate paragraph related to limitations of the study (page 16, lines 379-386).

The issues are:

The generalizability of this specific somatic mutation occurrence is very low (only 3/135 cGVHD) thus the significance for cGVHD patients as a whole is small.

#Although the occurrence of *mTOR P2229R* mutation is very low in our cohort, we consider that it could still play an important role in these individual cases and support abnormal growth of observed T cell clones. Our findings share similarities with recent study in CAR T cell setting where it was shown that the mutation in the *TET2* gene led to the expansion of single CAR T cell in a patient with chronic lymphocyte leukemia and enhanced therapeutic efficacy (Fraieta, Nature 2018).

In addition, somatic mutations may also exist in other genes and cell subsets, warranting further studies to elucidate the extend of this phenomenon in the setting of alloimmunity. However, we have revised the text and clarified both in abstract, results and discussion that we report studies of an index patient with cGVHD carrying persistent CD4+ T cell clonal expansion. Further, we state that "While somatic mutations in clonally expanded T cells in transplant recipients suffering from cGvHD is a novel observation, the generalizability of this phenomenon in cGvHD remains to be established by further studies" (page 16, lines 379-381).

Second, the patients and the control cohorts are extremely heterogeneous, for the type, age, cGVHD status, treatment status, time points of analyses, lack of donor analyses, the places of transplant- all of which make the conclusion that these mutations may be valid markers for cGVHD not particularly robust. Would suggest toning down of any discussion/ implication as a potential for cGVHD patients in the title and throughout the manuscript.

#We agree with the reviewer that this is heterogeneous cohort, but we feel that it represents well the clinical situation in cGvHD. We agree that our results cannot be generalized to all cGvHD patients and that was not our purpose. However, we consider that it is important to discover novel small subgroups of patients with individual pathogenesis. We have also added to the discussion that "Sequential samples starting from graft are also needed to understand the time of mutation formation and clonal evolution in detail" (page 16, lines 384-386). Further, we have revised the whole manuscript text and title as stated above.

The value of the other two mutations (*NFkB2* and *TLR2*), even though mentioned in abstract, introduction and esp *NFkB* in discussion, is neither clear nor validated. Would recommend tempering the implications of those mutations throughout the manuscript. In a similar note- the discussion throughout is extremely speculative including (not limited to the *DUSP2*, *NFkB*) and their implications. Would need to appropriately reduce and provide a more balanced context and content.

#We thank the reviewer for the suggestion and have now significantly enhanced the functional studies describing the effects of the *mTOR* mutation. In addition, we also modified the discussion and toned down the *NFkB2* and *TLR2* mutation parts as suggested. The functional studies for *NFkB2* and *TLR2* have been moved to the supplementary Figure S6.

cGVHD is complex and manifestations and treatments myriad from patient to patient and across physicians and centers. Please provide in greater detail the index and the other cGVHD patients

clinical profile and treatment context in greater detail. The index patient- was the person on sirolimus anytime before or during or after the time points of analysis. If cGVHD intensity increase, wax or wane or was it stable for the 2 years? If so how would the authors interpret the putative causal role of the CD4 clone to the changes in the disease? More details and also discussion on the level of immunosuppression (types and intensity of various drugs including steroids, CNIs etc) should be provided.

#We have summarized the patient characteristics in supplemental table S6. We have also added more details of the index patient and two other patients carrying *mTOR* mutations to the supplementary result. None of these patients with *mTOR* mutation were on sirolimus or everolimus before or after the sampling.

The index patient 1 had the most complicated history (summarized in supplementary figure S1) with multiple exacerbations of cGVHD (skin, liver, eyes) and subsequent titration of immunosuppressive regimen. *mTOR* mutation has persisted throughout the treatment and patient has still active cGVHD. In the other two patients (with *mTOR* mutation), the detection of the mutation matched with the development of cGVHD. In these two cases, cGVHD was successfully treated with immunosuppression. In the third patient with multiple samples (before, during and after cGVHD), the *mTOR* mutation was only detected at the onset of cGVHD (before initiation of immunosuppressive therapy), but not in the follow up samples.

We have previously shown in LGL leukemia that immunosuppressive treatment is not able to eradicate *STAT3* mutated LGL cells although some improvement in patients' disease status occurs (Rajala et al, Haematologica 2015). Thus, immunosuppressive drugs may partially inhibit the function of the mutated T cells and in this manner alleviate disease symptoms, but they are not able to eradicate the clone. Similarly, here with this cGVHD patient immunosuppressive drugs may partially improve the condition by inhibiting T cell function. And as also stated by the reviewer, the pathogenesis of cGVHD is multifactorial and used drugs may also affect the function of other cells involved in the process.

For the CTL assays with CD4+ T cells- is provocative. What were the negative target controls for the CD4 T cell clones?

#We agree that this cytotoxic CD4+ T cell clone is an interesting and provocative finding. We have now performed additional experiments to strengthen the data. As controls we have now added data from 3 healthy volunteers and cultured fibroblasts from the skin biopsy samples and performed the cytotoxicity assays with sorted CD4+ and CD8+ T cells from the healthy controls. We confirmed that CD4+ and CD8+ T cells from healthy controls did not kill their own fibroblasts. Second, we performed the cytotoxicity assay with multiple effector-target ratios and demonstrated dose-dependent cytotoxic effect CD4+ T cells, whereas similar effect was not noted with CD8+ T cells. Lastly, we hypothesized that the cytotoxicity of CD4+ T cells in the index patient is due to the recognition of MHC class I. In general, CD4+ T cells recognize MHC class II and fibroblasts should not have MHC class II. However, it has been reported that CD4+ T cells may also recognize MHC class I (PMID: 7650386, PMID: 15096184, PMID: 11797095). To test this, we used MHC class I and class II blocking antibodies. We confirmed that the cytotoxicity of CD4+ T cells was effectively blocked with MHC class I blocking, but not with MHC II blocking. These new data are now presented in Figure 5 and supplementary Figure S9.

Minor:

Some of the references do not seem to match the place where cited- too many to be specific, so please cross-check them all.

#We have checked all the references as suggested.

Reviewer #2, expert in T cell repertoire (Remarks to the Author):

The paper by Park and coworkers describes somatic mutations in a chronic GvHD patient potentially related to the GvHD. The analysis is impressive and the hypothesis regarding acquired somatic mutations driving cGvHD is provocative.

#We thank the reviewer for the positive feedback on our study.

Major comments

A major drawback is that the findings presented are confined to a single patient, making it however a very thorough case report. Only the identified mutations, but not their effect, are evaluated in a larger number of patients. Nonetheless the authors extrapolate to all cGvHD patients. The report is a case report at heart, and the broader value is uncertain given the prevalence of the mutations in cGvHD patients.

The findings are interesting, but more work (samples) is needed to elevate the report above anecdotal findings.

#Although the occurrence of *mTOR P2229R* mutation is very low in our cohort, we consider that it could still play an important role in these individual cases and support abnormal growth of observed T cell clones. We have added more functional data to the manuscript regarding the effects of *mTOR* mutation (Fig. 3 and supplementary Fig. S4). Our findings share similarities with recent study in CAR T cell setting where it was shown that the mutation in the *TET2* gene led to the expansion of single CAR T cell in a patient with chronic lymphocyte leukemia and enhanced therapeutic efficacy (Fraieta, Nature 2018).

However, we have now revised the text and clarified both in abstract, results and discussion that we report studies of an index patient with cGvHD carrying persistent CD4⁺ T cell clonal expansion. Further, we state that "While somatic mutations in clonally expanded T cells in transplant recipients suffering from cGvHD is a novel observation, the generalizability of this phenomenon in cGvHD remains to be established by further studies" (page 16, lines 379-381).

In terms of occurrence the TLR2 mutation seems much more interesting than the mTOR - the focus of the manuscript.

#We agree with the reviewer that *TLR2* mutation could be an interesting finding. However, as the mutation was also found in healthy controls' CD4⁺ T cells we consider that it is not cGvHD specific finding. In the revised manuscript based also on the comments by reviewer 1, we have modified the text and have focused on the effects of *mTOR* mutation since it was found only from cGvHD patients.

However, the true merit of the paper lies in identification of the CD4⁺ T-cells in cGvHD as being highly cytotoxic. This should be evaluated for the remaining 134 patients, the non-cGvHD patients and the healthy controls. The real-time cytotoxic assay is attractive for its functionality, but should be followed up by gene analysis where real-time PCR on bulk CD4 and CD8 will suffice.

#Although we agree that the finding of cytotoxic T cells is highly interesting, our main aim was to understand if somatic mutations can be detected in expanded T cell clones and whether they could explain the aberrant persisting T cell proliferation. However, we have now also performed new experiments regarding the cytotoxic function of CD4⁺ T cells. Unfortunately, we did not have suitable samples available from this large cohort, but from a small subset of patients and healthy controls (in total 17) we studied the potential cytotoxic nature of CD4⁺ T cells by granzyme B (GrB) flow cytometry analysis. Interestingly, our results supported the finding from the index patient and showed that the large proportion of CD4⁺ T cells in GvHD patients are GrB positive, whereas in healthy controls this is very rare. Furthermore, T cells from the patients showed increased Th1 cytokine production (TNF α and IFN γ) although the patients have been treated with immunosuppressive drugs. This new data is now presented in Figure 1f and the supplementary Figure S8.

Given higher prevalence of the TLR2 mutation in cGvHD than in healthy controls, an interesting question is if this mutation is associated with, or even predictive for, cGvHD.

#We agree that *TLR2* mutation is an interesting finding. However, as this is not germline event, and we do not have samples available before the allo-HSCT, no predictions of the development of GvHD can be done based on these results at the moment.

The abstract makes the reader believe the body of work is carried out on a cohort of 135 patients - this is not the case. The title is also misleading and the constant extrapolation from one patient to all cGvHD patients is exhausting.

#We have revised the text and clarified now both in abstract, results and discussion that we report studies of an index patient with cGvHD carrying persistent CD4⁺ T cell clonal expansion. Further, we have revised the title and concluded in the discussion that the generalizability of these results in cGvHD remains to be established by further studies (page 16, lines 379-386).

The presented evidence does not support the notion that the mutations were acquired after the transplantation: The authors compare mutations in a T cell population with clearly expanded clones to a population of unknown clone distribution, making it possible that the original clone with the mutation is in such a low frequency it avoids detection in the donor. The apparent increase in the cells harboring the mutations in at least mTOR seems to be increasing with time.

#We agree with the reviewer that it is not possible to totally rule out that the mutation was initially in the HSCT-donor. However, we have sorted donor CD4⁺ and CD8⁺ T cells and studied them with ultra-deep sequencing (total coverage 700 000) and no *mTOR* mutation was discovered (data added to the results, page 6, lines 147-150). In the discussion we have added the sentence stating "Sequential samples starting from graft are needed to understand the time of mutation formation and clonal evolution in detail" (page 16, lines 384-386). Further, we have also added new sequencing data from patient samples and shown how mutation VAF increases in CD4⁺ T cells during 6 years of follow-up period (Fig 2c and Supplementary table S3).

Do the T-cells using the Vb20 also increase over time as does the mTOR mutation?

#We thank you for the comment and have now included the data showing how the Vβ20 bearing clonotype increases over time in Figure 1c.

Do other cGvHD patients have expanded clonotypes?

#It has been reported earlier that cGvHD patients show clonal T cell expansions (PMID: 12379890). This is also referred in the manuscript (Ref 5-7).

7. The mTOR P2229R mutation is found in 2.2% of the evaluated cGvHD patients. What about the remaining 97.8% - do they have other mutations in mTOR or TLR2? Do the three patients have other things in common other than the mTOR P2229R mutation?

#We agree that this is an interesting question which requires further studies. To address this properly, new sample collection starting already from the graft is needed. In addition, from our validation cohort we do not have suitable samples available (sorted CD4⁺ and CD8⁺ T cells) which would allow discovery of novel mutations in T cells. We have revised the text as described above and agree that further studies are needed to address the full landscape of somatic mutations in lymphoid cells in patients with cGvHD. The clinical characteristics of 3 cases are now described in detail in the supplementary results. All cases had cGvHD (not acute GvHD), but otherwise their disease histories differed.

The pooling of TCRs - where the beta-chain or alpha-chain were missing, but the present counterpart had identical CDR3 to an expanded TCRab set - assuming identical source is problematic. Is there further evidence that the events should be pooled - other than the CDR3?

#Based on this comment, we redefined the pooling of the mutated clonotype to include only two of the biggest clonotypes that had the same Vβ, CDR3b and Jb genes. We also have TCRβ data

from V β 20 sorted clonotype that has the same V β , CDR3b and J β genes as the pooled clonotypes in the single-cell assay.

The TCR nomenclature should be unified. The 'IOtest Beta Mark TCR V β Repertoire Kit' makes use of a variation of the Arden nomenclature, while Adaptive Biotechnologies have their own variation of the IMGT nomenclature. I suggest sticking with the IMGT nomenclature, but any one is possible as long as it is coherent and clearly stated in the materials and methods.

#We thank the reviewer for the suggestion. The corresponding IMGT gene nomenclature for V β 20 (the nomenclature from Wei et al., PMID: 8206523) in IOtest Beta Mark TCR V β Repertoire Kit is TRBV30. As suggested by the reviewer, we have now noted IMGT gene nomenclature in the methods. Additionally, we have provided the corresponding IMGT gene nomenclature in the nomenclature used for V β segment in the kit in the supplementary table S8 to be precise.

The presentation of the discovery of the mTOR, NF κ B2, and TLR2 mutation and their accumulation is confusing. Their association to V β 20 is also not made clear. As this is a supporting pillar of the manuscript it should be rectified.

#We have revised the figures and the text to clarify these parts. Mutations have been discovered in multiple timepoints during 6 years follow-up. We have added to the figures both the change in V β 20 clone (Fig. 1c) and mutation VAF (Fig. 2c) over time. In addition, we have sorted the V β 20 positive cells and shown that the mutations exist in this fraction (page 6, lines 139-143, table S2).

Figure 3: It is not necessary to evaluate the effect of the NF κ B2 mutation given that this mutation is confined to the index patient. It would be great to see the effects of the mTOR mutation alone, the TLR2 mutation alone, and the two combined. At least the individual effects of the mutations should be demonstrated and not just the combined effect.

#We thank the reviewer for the suggestion. We consider NF κ B2 mutation is still important since it is a crucial gene for T cell function. However, we have now significantly enhanced the functional studies focusing on mTOR mutation since the mTOR mutation was commonly found from 3 patients. We have individually analyzed the effects of all three different mutations. The new data is presented in Figure 3. We have also moved the functional data showing HEK293 cells expressing NF κ B2 mutation and TLR2 mutation to the supplementary Figure S6. In addition, we toned down the discussion about NF κ B2 and TLR2 in the manuscript.

Figure 4:

It is unclear what is being presented exactly. CD4+ cells from 2015 and 2017 were sequenced, but only one set of data is shown.

#As we wanted to address how similar cells from two different time points are, we sequenced samples separately, but analyzed them together. In the figure legend we state that the figure 4 contains pooled data from both of the time points.

The supplement figure S5 show a less than 2 fold change in overall gene expression between the two time points as argument for pooling, but afterwards the mTOR mutation. Given an increase in VAF of the mTOR mutation over time, the two samples should be split - at least to demonstrate that they do not differ dramatically and can be pooled for further analysis.

In figure 4, both of the samples (2015 and 2017) are shown as pooled. We did not notice any major differences between these two data based on multiple criteria. First, in dimensional reduction analyses, the cells from two different samples merge well with each other and no granularity can be seen. This is added as supplementary Fig. S7. Second, the fold-change analysis shows that in our re-analysis only the cycling-cell population increases over two-fold, meaning that there is no dramatical difference. The increase in VAF of the mTOR mutation is only 1% between these two time-points (Fig. 2c) and therefore, no major changes in the phenotype are observed.

The criteria for 'similar phenotype' used in D and E are elusive and should be clearly defined.

#We revised this part and the separation should be now clearer, as we are comparing cluster against cluster.

It is extremely surprising that the majority CD4+ T-cells form a cluster which can be termed cytotoxic. When zooming into figure 4B it becomes evident that among others granzymes and perforin help to define the clusters as cytotoxic. Only traces of CD8 was found in all clusters indicating that a vast majority of the CD4 T-cells in this cGvHD patient are cytotoxic. The authors, however, does not seem to be baffled by this extremely counter-intuitive observation. More thoughts and observations devoted to this observation is needed.

We agree that this is an interesting finding. We have added analysis of single cell data from a healthy donor (Fig. 4d), where we show that these cytotoxic markers cannot be found with the same single-cell technology and analysis in a healthy donor (page 10, lines 233-235). Further, we have done flow cytometry analysis using granzyme B as a marker for cytotoxic cells and show that cGvHD patients have markedly increased amount of these cells compared to healthy controls (Supplementary Fig. S8).

A $\log_2(\text{FC})$ of 0.25 is not dramatic (1.2 times more in one compared to the other population) and the expression of only two genes is half or double that of the reference population. To verify this is indeed associated with the Vb20 mTOR-mut sub-population and above the noise level, the observations should be compared to repeat random sampling of all the cells in the cytotoxic cluster.

#We thank the reviewer for the comment. We would like to emphasize that it is not $\log_2(\text{FC})$, but rather average $\log_2(\text{FC})$ of the single cells in the populations. In single-cell transcriptomics analyses, unlike in bulk, the reported fold-changes are usually average foldchanges across the pools. Thus, the fold-changes are constantly less than they are in bulk, due to dropouts and differences in the distribution models, and more emphasis should be put on the p-value from the statistical analysis.

<Minor comments>

I suggest the use of the word 'AIRR-seq' (adaptive immune receptor repertoire) or 'TCR AIRR-seq' in place of TCRab-seq.

#We thank you for the comment. However, if possible, we would prefer to keep the term TCR $\alpha\beta$ -seq as it is commonly used in translational papers and not all readers are very familiar with the other term and could be confused. However, if considered necessary, we can revise the terminology as suggested.

It would be interesting to see plots for NFkB2 and TLR2 similar to mTOR in figure 2C, where the increase of mutation burden is evident. Given that also other mutations were identified in CD4+, their development over time would also be interesting to see.

#We appreciate the suggestion. We have added the plots for *NFkB2* and *TLR2* as well as for *mTOR* in Figure 2c.

There are quite a few imprecisions and grammatical errors - a few are mentioned below though the list is not exhaustive. The two chains of the TCR are termed α and β , please stick to this notation and avoid Vb, TCRa, TCRb, and TCRab

#We apologize for the grammatical errors and revised them including α and β .

The identified expanded Vb20 bearing clonotype is usually referred to as 'the clone' or 'the clonotype'. A more precise notation in in order and could be 'Vb20 bearing clonotype' though other possibilities also exist

#We apologize for the non-precise terming and changed to V β 20 bearing clonotype.

The discussion skips on the ongoing debate of cytotoxic CD4+ T cells in GvHD. One paper that comes to mind is Du et al, J Immunol., 2015 (doi: 10.4049/jimmunol.1500668) but there are others equally relevant.

#We have added this paper to the discussion (page 16, lines 365-366). In addition, we have added

new data of cytotoxic CD4⁺ T cells both in index patient and other cGvHD patients (Figure 5, Supplementary Fig. S8).

Given that some plots are non-Graphpad Prism, the appropriate sources should be cited
#We have added all sources that we used in the figure legends and in methods.

<Abstract>

Lines 51-52: This is a clear overstatement and should be toned down dramatically
#We have revised the whole manuscript considerably and also this sentence has been revised.

<Introduction>

Line 59: Give frequency rather than just 'in many patients'
#Thank you for the comment, we have modified it in detail (page 3, line 57-58).

Line 61: Please insert reference (not for the organs, but for the lymphocytes bearing the driver)
#We have revised the sentence and added reference as suggested (page 3, line 62).

Line 69-70: An overstatement; the mechanism - host derived antigen - is clear and eloquently described in the beginning of the paragraph.
#We have removed this sentence and revised the text.

<Results>

Line 91: Given that only the V-gene usage is assessed, it is misleading to speak of clones. More importantly, 'clonality' is a very loosely defined concept within T-cell biology owing to the complete lack of a universal formal definition and should always be avoided; or at the least formally defined.
#We have revised the terms in the manuscript and aimed to avoid term clone (page 4, line 90).

Line 94: To the best of my ability, I cannot see Vb20 in CD8⁺ in Fig 1B. The purpose of Fig 1A is elusive since TCR V-beta for CD8 is not shown. The purpose of showing Total CD4 and CD4⁺CD8⁺ without further mention is equally elusive.
#We apologize the elusive explanation in the result and the figure legend. First, the aim of Fig. 1a is to show the gating strategy how we separated the population of CD4⁺ and CD8⁺ T cells from whole blood to screen all TCR Vβ antibodies. From the screening we confirmed that CD4⁺ T cells have large clonal Vβ20 expansion, whereas it was not observed among CD8⁺ T cells (Fig. 1a). We have modified that the Fig. 1a (middle panel and right panel) present Vβ20 from CD4⁺ T cells and CD8⁺ T cells. In addition, we have added TCR V-beta for CD8⁺ T cells in Supplementary Figure S2c.

Line 96: See comment for Line 91
#We have revised the text as suggested (page 4, line 95).

Line 100: How do these two V-genes translate to the flow cytometry results?
#TCRBV07-09 detected by TCRβ sequencing of CD8⁺ T cells does not have the corresponding Vb antibody in the flow panel. TCRBV28-01 should correspond Vb3 in flow, but for some reason we were unfortunately not able to detect it by flow cytometry staining. TCRBV30-01 expansion in CD4⁺ T cells (corresponding to the Vβ20 antibody staining in flow) was confirmed both by flow cytometry and single cell sequencing.

Lines 101-108: The value of the information is unclear. There is no information to extract from a comparison to the HSCT-donor, rather a before-after the flare comparison would be more appropriate. The value of such information in the context of the paper remain unclear
#We thank you for the comment. With this analysis, we wanted to show the basic phenotype of patient cells. Unfortunately, we do not have samples taken before and after the flare. We have now added to the figure data of granzyme B positive cells. However, if reviewer prefers, we can also

transfer these to supplemental figures.

Line 144: I fail to find the VAF information behind figure 2C - I am only able to find CD4+ from 2015 in Table 2. The tabular summaries are great. Given that there was refinement of the question over time, it would be nice to see the summaries for all compared samples (for instance CD4+ from 2013, 2015, and 2017 in the same table)

#We have now added this information to the table 2 as suggested.

Line 146: Incorrect figure reference

#We apologize that the sentence may be confusing. We have revised the sentence and checked that it is referring correct figures (page 6, line 141).

Lines 153-158: Are Vb20 CD4 T-cells only present in skin lesions? Maybe the mTOR mutation is inconsequential and their expansion is due to antigen in the skin.

#Unfortunately, we do not have samples available any longer from which we could address if the V β 20 cells are only observed in skin.

Line 166: Indication that the index is part of the 135 patient cohort. If the total number of all patients with cGvHD 135 or is it 136? If it is 135, it is more prudent to use indicate the validation cohort to be of 134 cGvHD patients

#The total number of cases are 135 including index patient and have modified the text accordingly (page 7, line 162).

<Materials and methods>

The index patient is not mentioned. Was the patient in a separate cohort?

#Thank you for the comment. We have modified the section of study patients.

Line 375: Please add 'allo-HSCT'

#We thank for the comment and have added 'allo-HSCT' as requested.

Line 392-393: Missing clone and cat#

#We have added the cat# for the secondary antibodies. In case of the primary antibodies, cell signaling does not assign clone IDs for the polyclonal antibodies unless they want to differentiate between polyclonal antibodies against the same target. Thus, NFkB2 (Cat#: 4882S, Lot: 4), phospho-Akt (Ser473) (Cat#: 9271T, Lot#: 14), 4E-BP1 (Cat#: 9452, Lot#: 12), phospho-FoxO1 (Thr24)/FoxO3a (Thr32) (Cat#: 9464, Lot#: 7), phospho-p70S6 (S6K) kinase (Thr421/Ser424) (Cat#: 9204S, Lot#: 11), and polyclonal antibody Deptor (Cat# NBP-149674SS, Lot: D2, Novus Biologicals) do not have clone IDs.

Line 399: Duplicate and misleading information; by reading it becomes clear that HEK293 and not HEK293FT were verified

#We thank you for the comment and modified more clearly with HEK293FT authentication.

Line 400: Missing preposition

#We thank for the comment and revised it.

Line 427 and elsewhere: FACS not FACs

#We thank for the comment and modified all.

Lines 480-496: It is not clear that the analysis is performed in R, it is not clear how the dimensionality is reduced. Generally, it is hard to follow the analysis.

#We thank for the comment and have revised the text to be more detailed.

Line 502-504: The first sentence is nonsensical. It is not clear what 'two recombinations' entail and

hints that commonly occurring T-cells using the same beta-chain with two different alpha-chains are excluded.

#We revised the text to be more clear.

Line 516: Should probably read 200-300 μ l

#We thank for the comment and revised it (page 25, line 577).

Line 517: Switch from past to present tense

#We thank for this comment. It was present tense and now we have revised it to past tense (page 25, line 579).

Line 528: Maybe write as a proper equation

#We thank for the comment and added the formula (page 26, line 594-603).

Lines 524-534: It is unclear how long the fibroblasts were allowed to settle

#In the sentence "100 μ l of the cell suspension (8×10^3 cells/well) was transferred to the plate followed by incubation at room temperature for 30 min to allow the cells to settle at the bottom of the well", the cells mean the fibroblasts. We thank the reviewer for the comment and have modified the word from the cells to fibroblasts. Additionally, we revised the "Analysis of cellular cytotoxicity" part.

Line 605: It is not clear if cells from two or three donors were used in the screen

#We thank you for the comment and have added more details to be clear. Basically, CD4⁺ T cells were obtained from a healthy donor, a HSCT donor who is the sibling of the index patient, and from the index patient. We have put the word, "HSCT donor" instead of just "donor" for clear explanation.

<Discussion>

Lines 262-263: Clear overstatement. It is really hard to extrapolate from 2.2% to all cGvHD patients.

#We have re-written the discussion and this sentence has been removed.

Line 263: That the mTOR mutation is limited to the expanded V β 20 T-cell clone is not formally shown.

#We have sorted various cell populations and analyzed them with amplicon sequencing. In the CD4⁺ fraction the VAF of the mutation was: 19.2% whereas in the sorted CD4⁺V β 20+ population the VAF was 44.7%. As the size of the V β 20 population from all CD4⁺ cells was around 50%, the VAF values match very well that the mutation is confined to V β 20 population.

<Figures and tables>

Table 1: Header: It is a minor annoyance to see the word 'tumour' in relation to the analysis of bulk CD4+

#We have corrected it and replaced the word tumor with CD4.

Line 859: 'expanded CD4+' is misleading. According to the material and methods as well as the main text it is all CD4+ T cells that were evaluated.

#We thank for the reviewer's comment and have modified it.

Figure 1A: The plot annotation is too small to be read. Unclear what TCR V-beta FITC and TCR V-beta PE represent and this is not explained in the figure caption.

#We thank you for the comment and we have revised the figure and figure legend.

Figure 1B: The axis labels and titles are too small to be read. The plot would benefit from a clear indication of the information (CD4 or CD8) as a title, in the axis title, or both

#We apologize for the small text in labels and titles. We now revised the axis labels and titles as suggested.

Figure 1C: The axis labels and titles are too small to be read. 3D plots should be shunned, and could be represented in a heatmap or - given the few number z-axis values - in individual plots
#We would like to use the 3D plot for the result, as we feel that this quite well presented the TCR repertoire and similar presentation has been used in many other TCR publications. However, we agree that text was difficult to read, and we have modified the figure.

Figure 1D: The plot annotation is too small to be read.
#We changed the plot annotation as suggested. The data is presented in Figure 1e.

Figure 2B: The nucleotide calls are too small to be read.
#We thank for the comment and revised the figure.

Figure 2D: The colouring is too weak to be seen clearly and the indicator arrows are white blobs - especially in a print out. What are the two images? The caption gives no hint. From the main text it becomes clear that the aim is to underline T-cell infiltration in the skin lesion. Can this be quantified? Can this be compared to infiltration in the affected eye and liver, and directly compared with the VAF? As it is now, the information is in three places: text, table and figure. If possible it should be combined
#We apologize for the unclear figure and have changed the figure to include more clear pictures and put the titles in detail. The *mTOR* mutation was only observed in skin biopsy and we have now added a graph showing VAFs from the index patient's skin, liver, and eyes biopsy in Figure 2d.

Figure 3B-3D: The axis labels and titles are too small to be read. The bars indicate SEM. Unless each point is the mean of a number of samples (in which case the numbers should be added), SEM is the wrong statistics and should be replaced by SD
#We thank the reviewer and agree that the statistics should be done by SD. We apologize that we made a mistake when we wrote that they are SEM even though we used the statistics with SD. On the whole we modified figure 3 focusing on *mTOR* mutation in the revised manuscript.

Figure 4: All axis labels and titles are too small to be read.
#We have revised the figure as suggested.

Figure 4D: The plot make use of two different red, but the caption mention red and gray. There should be a legend on the plot
#We apologize for the confusion and have now revised the figure in Fig. 4e.

Figure 4E: The non-red colour is practically invisible when printed. What does that colour represent? There should be a legend on the plot. Does the y-axis really represent log 100 of the p-value?
#The y-axis represents log100 of the p-value. In single-cell analyses, the p-values are astronomically small and, in some cases, approaches 0. We have now transferred the axis to log10. We have modified the figure and the legend in Fig. 4f. The red color represents the mutated *mTOR* clonotype.

Figure 4E: The resolution is too low and the annotation too small
#We thank for the comment and revised the figure in Fig. 4f.

Figure 6: All axis labels and titles are too small to be read.
#We have revised the figure as suggested.

Figure 6B; 6D: axis title and labels are missing for the x-axis
#We now added the title for x-axis. In addition, the labels for drugs were magnified.

Figure 6F; 6G: See comment for Figure 3B-3D

#This has now been corrected as mentioned for Figure 3B-3D and has moved to the supplementary Fig. S11.

<Supplementary>

Page numbers missing in table of contents

#We have added all page numbers.

Figure S2: What is 'donor' exactly? Only by reading the main text does it become clear that it is the HSCT-donor and not some healthy blood donor

#We thank the reviewer for this comment and have modified to "HSCT donor". In addition, we removed Fig. S2 since it was same with Fig. 1e.

Figure S3: Different symbols are fine, but please use x-axis labels.

#We have added the x-axis labels.

Figure S4A: The evaluation of expression levels should be shown for all introduced constructs.

#We apologize for the unclear figure legend and have modified the legend by adding the exact number for all introduced constructs in Fig. S6c.

Figure S4B: This is relevant enough to be in the main figure

#Based on comments by other reviewers we now focused mostly on *MTOR* mutation and have modified the figures accordingly. Also, the *NFkB2* data has been moved from the main figure to supplementary figure (Supplementary Fig. S6a - S6b).

Primer sequences of Table S3, S5, S6 cannot be extracted directly, they should be readily available.

#Thank you for the comment. We have modified them (Supplementary Table S9, S11, and S12).

Reviewer #3, expert in mTOR pathway (Remarks to the Author):

The authors identified persisting somatic MTOR, NFKB2, and TLR2 mutations in an expanded CD4+ T clone. The MTOR P2229R kinase domain mutation was detected in two additional cGvHD patients, but not in controls. Functional analysis of the discovered MTOR mutation indicated a gain-of-function alteration in translational regulation yielding in up-regulation of phosphorylated S6K1, S6, and AKT. Paired single-cell RNA and TCR sequencing revealed cytotoxicity and abnormal proliferation of the clonally expanded CD4+ T cells. The results are interesting but need additional support.

#We thank the reviewer for the positive feedback on our study and have added new data for additional support as detailed below.

Most importantly, the authors need to prove that MTOR P2229R mutation is a truly gain-of-function by introducing such mutation into wild-type cells and test the functional outcome (also see below). Without such data, the observed phenotypes could be entirely secondary to alterations in other genes or pathways.

#We agree with the reviewer that we need to validate the gain-of-function of *mTOR P2229R* mutation in more detail to better understand the potential functional significance of the mutation. In the original manuscript, we showed the data of phosphorylation of S6, S6K1, and Akt using HEK293 cells stably expressing *mTOR* wild-type (WT) and *mTOR P2229R* mutant. We have now significantly enhanced the functional study including new data from mTOR activation pathway (detailed below). Further, we have performed all experiments with single *mTOR*, single *TLR2* and single *NfkB2* mutants as well as all mutants combined to show that effects are related to *MTOR P2229R* mutation.

The authors need to perform additional assays related to MTOR signaling, including p-4EBPs, p-Foxo1, cell proliferation, cell death, cytokine production, etc.

#We have performed additional assays to show more in-depth the effect of *mTOR P2229R* mutation. First, we performed cell proliferation analysis from HEK293 expressing *mTOR* mutation and *mTOR WT* under the standard condition (Media including 10% FBS). Secondly, we also added apoptosis analysis by PI-Annexin V staining from flow cytometry under the serum-starvation condition. The data is shown in Figure 3a-c.

Next, we have studied additional significant proteins in mTOR pathway as suggested by the reviewer. We first checked highly expressed mTOR protein from HEK293 cells expressing *mTOR* WT and *mTOR P2229R* mutant. In addition to phospho-S6, S6K1, and Akt protein, we have provided the protein expression data for phospho-4EBP1, total 4EBP1, phospho-FoxO1/3, total FoxO1/3, TSC1, and TSC2. The data is presented in Figure 3d, and in the results section. Furthermore, we confirmed no protein expression change from HEK293 stably expressing *TLR2 WT*, *TLR2 W558L* mutation, *NfkB2 WT*, *NfkB2 P882Q* mutation, and Triple WT (*mTOR*, *TLR2*, and *NfkB2*) to support the gain-of-function effect of *mTOR* mutation.

Finally, the authors need to carefully analyze the effects of MTOR P2229R on mTORC1 and mTORC2 pathways.

#We thank the reviewer for the suggestion. To analyze the effect of *mTOR P2229R* mutation we showed the expression of p-4EBP1, p-S6K1, and p-S6 implicating mTORC1. Moreover, we have provided the protein expression of p-FoxO1/3, p-Akt, and TSC1/2 indicating mTORC2. As mentioned in previous comment, the data is presented in Figure 3d. Furthermore, we have now added additional data showing mTORC1 and mTORC2 specific binding proteins Raptor and Rictor, respectively. Given that we did not find the difference of Raptor and Rictor, we have performed immunoprecipitation assay to show the protein level of Deptor (a negative inhibitor of mTOR) as suggested in the previous study (PMID: 24631838). This supports mTORC1 and mTORC2 activation in *mTOR P2229R* mutation. The data is presented in Figure 3d-f, and the result section has been modified (page 8, line 174-223).

REVIEWERS' COMMENTS:

Reviewer #1 (Remarks to the Author):

The authors have been responsive and made appropriate changes to their interpretations throughout and show new experimental datasets for CD4 cytotoxicity. Overall well-performed and well-written. Acceptable.

Minor:

There are a few typos, that can be corrected.

Reviewer #2 (Remarks to the Author):

I went through the manuscript checking all critics points.
I feel that the points are adequately addressed

Reviewer #3 (Remarks to the Author):

The authors have largely addressed my questions. The western blot data presented in Fig. 3 and supplementary figure 6 need to be quantified for statistical analysis to compare the pathway activities between wild-type and MTOR P2229R mutation.

Response to the reviewers' comments:

Reviewer #1 (Remarks to the Author):

The authors have been responsive and made appropriate changes to their interpretations throughout and show new experimental datasets for CD4 cytotoxicity. Overall well-performed and well-written.

Acceptable.

Minor:

There are a few typos, that can be corrected.

#Thank you for the positive comments. We have checked the text and corrected all typos that we have noted.

Reviewer #2 (Remarks to the Author):

I went through the manuscript checking all critics points. I feel that the points are adequately addressed

#We thank the reviewer for the positive feedback.

Reviewer #3 (Remarks to the Author):

The authors have largely addressed my questions. The western blot data presented in Fig. 3 and supplementary figure 6 need to be quantified for statistical analysis to compare the pathway activities between wild-type and MTOR P2229R mutation.

#Thank you for the positive comments. We have quantified the western blot analysis and provide them as Supplementary Figures 4 and 5.